# Learning World Models with Identifiable Factorization

**Yu-Ren Liu**[*,1,4], **Biwei Huang** [*,2], **Zhengmao Zhu**[1,4], **Honglong Tian**[1],

**Mingming Gong**[5,4], **Yang Yu**[†,1,6,7], **Kun Zhang**[†,3,4]

[1] National Key Laboratory for Novel Software Technology, Nanjing University, China
[2] University of California San Diego, USA
[3] Carnegie Mellon University, USA
[4] Mohamed bin Zayed University of Artificial Intelligence, UAE
[5] University of Melbourne, Australia
[6] Polixir.ai, China
[7] Peng Cheng Laboratory, China

{liuyr,zhuzm, tianhl, yuy}@lamda.nju.edu.cn bih007@ucsd.edu
mingming.gong@unimelb.edu.au kunz1@cmu.edu

## Abstract

Extracting a stable and compact representation of the environment is crucial for efficient reinforcement learning in high-dimensional, noisy, and non-stationary environments. Different categories of information coexist in such environments – how to effectively extract and disentangle the information remains a challenging problem. In this paper, we propose IFactor, a general framework to model four distinct categories of latent state variables that capture various aspects of information within the RL system, based on their interactions with actions and rewards. Our analysis establishes block-wise identifiability of these latent variables, which not only provides a stable and compact representation but also discloses that all reward-relevant factors are significant for policy learning. We further present a practical approach to learning the world model with identifiable blocks, ensuring the removal of redundancies but retaining minimal and sufficient information for policy optimization. Experiments in synthetic worlds demonstrate that our method accurately identifies the ground-truth latent variables, substantiating our theoretical findings. Moreover, experiments in variants of the DeepMind Control Suite and RoboDesk showcase the superior performance of our approach over baselines.

## 1 Introduction

Humans excel at extracting various categories of information from complex environments [1, 2]. By effectively distinguishing between task-relevant information and noise, humans can learn efficiently and avoid distractions. Similarly, in the context of reinforcement learning, it's crucial for an agent to precisely extract information from high-dimensional, noisy, and non-stationary environments.

World models [3, 4] tackle this challenge by learning compact representations from images and modeling dynamics using low-dimensional features. Recent research has demonstrated that learning policies through latent imagination in world models significantly enhances sample efficiency [5, 6, 7, 8]. However, these approaches often treat all information as an undifferentiated whole, leaving policies susceptible to irrelevant distractions [9] and lacking transparency in decision-making [10].

This paper investigates the extraction and disentanglement of diverse information types within an environment. To address this, we tackle two fundamental questions: 1) How can we establish a

---

[*]: Equal contribution. [†]: Corresponding authors.

37th Conference on Neural Information Processing Systems (NeurIPS 2023).

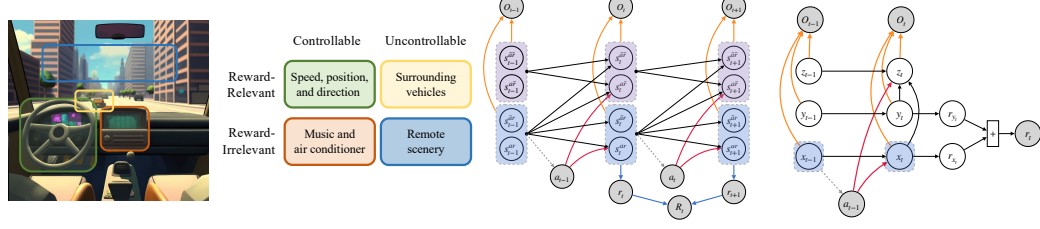

| | Controllable | Uncontrollable |
| --- | --- | --- |
| Reward-Relevant | Speed, position, and direction | Surrounding vehicles |
| Reward-Irrelevant | Music and air conditioner | Remote scenery |

(a) Car-driving task.    (b) Four types of states.    (c) IFactor (Ours).    (d) Denoised MDP.

Figure 1: (a) An illustrative example of the car-driving task. (b) Four categories of latent state variables in the car driving task. (c) The structure of our world model. Grey nodes denote observed variables and other nodes are unobserved. We allow causally-related latent processes for four types of latent variables and prove that they are respectively identifiable up to block-wise invertible transformations. We show that both $s_t^{ar}$ and $s_t^{\bar{a}r}$ are essential for policy optimization. The inclusion of the gray dashed line from $(s_t^{ar}, s_t^{\bar{a}r})$ to $a_t$ signifies that the action could be determined by reward-relevant variables. (d) The structure of Denoised MDP [9]. It assumes the latent processes of $x_t$ and $y_t$ are independent and uses only $x_t$ for policy optimization. The gray dashed line from $x_{t-1}$ to $a_{t-1}$ shows that the action could be determined only based on the controllable and reward-relevant latent variables. Further, the existence of instantaneous causal effect from $x_t$ and $y_t$ to $z_t$ renders the latent process unidentifiable without extra intervention on latent states [14].

comprehensive classification system for different information categories in diverse decision scenarios? 2) Can the latent state variables, classified according to this system, be accurately identified? Prior research has made progress in answering the first question. Task Informed Abstractions [11] partition the state space into reward-relevant and reward-irrelevant features, assuming independent latent processes for each category. Iso-Dream [12] learns controllable and noncontrollable sources of spatiotemporal changes on isolated state transition branches. Denoised MDP [9] further decomposes reward-relevant states into controllable and uncontrollable components, also assuming independent latent processes. However, the assumption of independent latent processes is overly restrictive and may lead to decomposition degradation in many scenarios. Moreover, none of these approaches guarantee the identifiability of representations, potentially leading to inaccurate recovery of the underlying latent variables. While discussions on the identifiability of representations in reinforcement learning (RL) exist under linear assumptions [13], the question of identifiability remains unexplored for general nonlinear cases.

In this paper, we present IFactor, a general framework to model four distinct categories of latent state variables within the RL system. These variables capture different aspects of information based on their interactions with actions and rewards, providing transparent representations of the following aspects: (i) reward-relevant and controllable parts, (ii) reward-irrelevant but uncontrollable parts, (iii) controllable but reward-irrelevant parts, and (iv) unrelated noise (see Section 2). Diverging from prior methods, our approach employs a general factorization for the latent state variables, allowing for causally-related latent processes. We theoretically establish the block-wise identifiability of these four variable categories in general nonlinear cases under weak and realistic assumptions. Our findings challenge the conclusion drawn in the Denoised MDP [9] that only controllable and reward-relevant variables are required for policy optimization. We emphasize the necessity of considering states that directly or indirectly influence the reward during the decision-making process, irrespective of their controllability. To learn these four categories of latent variables, we propose a principled approach that involves optimizing an evidence lower bound and integrating multiple novel mutual information constraints. Through simulations on synthetic data, we demonstrate the accurate identification of true latent variables, validating our theoretical findings. Furthermore, our method achieves state-of-the-art performance on variants of RoboDesk and DeepMind Control Suite.

## 2   Four Categories of (Latent) State Variables in RL

For generality, we consider tasks in the form of Partially Observed Markov Decision Process (POMDP) [15], which is described as $\mathcal{M} \triangleq (\mathcal{S}, \mathcal{A}, \Omega, R, T, O, \gamma)$, where $\mathcal{S}$ is the latent state space, $\mathcal{A}$ is the action space, $\Omega$ is the observation space, $R : S \times A \to \mathbb{R}$ defines the reward function, $T : S \times A \to \Delta(S)$ is the transition dynamics, $O : S \times A \to \Delta(\Omega)$ is the observation function, $\gamma \in [0, 1]$ is the discount factor. We use $\Delta(S)$ to denote the set of all distributions over $S$. The agent can interact with the environment to get sequences of observations $\{\langle o_t, a_t, r_t \rangle\}_{t=1}^{T}$. The objective is to find a policy acting based on history observations, that maximizes the expected cumulative (discounted) reward.

In the context of POMDPs, extracting a low-dimensional state representation from high-dimensional observations is crucial. Considering that action and reward information is fundamental to decision-making across various scenarios, we disentangle the latent variables in the environment into four distinct categories, represented as $\mathbf{s_t} = \{s_t^{ar}, s_t^{\bar{a}r}, s_t^{a\bar{r}}, s_t^{\bar{a}\bar{r}}\}$, based on their relationships with action and reward (see Figure 1(c) as a graphical illustration):

- Type 1: $s_t^{ar}$ has an incident edge from $a_{t-1}$, and there is a directed path from $s_t^{ar}$ to $r_t$.
- Type 2: $s_t^{\bar{a}r}$ has no incident edge from $a_{t-1}$, and there is a directed path from $s_t^{\bar{a}r}$ to $r_t$.
- Type 3: $s_t^{a\bar{r}}$ has an incident edge from $a_{t-1}$, and there is no directed path from $s_t^{a\bar{r}}$ to $r_t$.
- Type 4: $s_t^{\bar{a}\bar{r}}$ has no incident edge from $a_{t-1}$, and there is no directed path from $s_t^{\bar{a}\bar{r}}$ to $r_t$.

$s_t^{ar}$ represents controllable and reward-relevant state variables that are essential in various scenarios. Taking the example of a car driving context, $s_t^{ar}$ encompasses driving-related states like the current speed, position, and direction of the car. These latent variables play a critical role in determining the driving policy since actions such as steering and braking directly influence these states, which, in turn, have a direct impact on the reward received.

$s_t^{\bar{a}r}$ refers to reward-relevant state variables that are beyond our control. Despite not being directly controllable, these variables are still necessary for policy learning. In the context of car driving, $s_t^{\bar{a}r}$ includes factors like surrounding vehicles and weather conditions. Although we cannot directly control other cars, our car's status influences their behavior: if we attempt to cut in line unethically, the surrounding vehicles must decide whether to yield or block our behavior. As a result, we need to adjust our actions based on their reactions. Similarly, the driver must adapt their driving behavior to accommodate various weather conditions, even though the weather itself is uncontrollable.

The state variables denoted as $s_t^{a\bar{r}}$ consist of controllable but reward-irrelevant factors. Examples of $s_t^{a\bar{r}}$ could be the choice of music being played or the positioning of ornaments within the car. On the other hand, $s_t^{\bar{a}\bar{r}}$ represents uncontrollable and reward-irrelevant latent variables such as the remote scenery. Both $s_t^{a\bar{r}}$ and $s_t^{\bar{a}\bar{r}}$ do not have any impact on current or future rewards and are unrelated to policy optimization.

We next build a connection between the graph structure and statistical dependence between the variables in the RL system, so that the different types of state variables can be characterized from the data. To simplify symbol notation, we define $s_t^r := (s_t^{ar}, s_t^{\bar{a}r})$, $s_t^{\bar{r}} := (s_t^{a\bar{r}}, s_t^{\bar{a}\bar{r}})$, $s_t^a := (s_t^{a\bar{r}}, s_t^{ar})$ and $s_t^{\bar{a}} := (s_t^{\bar{a}r}, s_t^{\bar{a}\bar{r}})$. Specifically, the following proposition shows that $s_t^r$ that has directed paths to $r_{t+\tau}$ (for $\tau > 0$), is minimally sufficient for policy learning that aims to maximize the future reward and can be characterized by conditional dependence with the cumulative reward variable $R_t = \sum_{t'=t} \gamma^{t'-t} r_{t'}$, which has also been shown in [13].

**Proposition 1.** *Under the assumption that the graphical representation, corresponding to the environment model, is Markov and faithful to the measured data, $s_t^r \subseteq \mathbf{s_t}$ is a minimal subset of state dimensions that are sufficient for policy learning, and $s_{i,t} \in s_t^r$ if and only if $s_{i,t} \not\perp\!\!\!\perp R_t | a_{t-1:t}, s_{t-1}^r$.*

Moreover, the proposition below shows that $s_t^a$, that has a directed edge from $a_{t-1}$, can be directly controlled by actions and can be characterized by conditional dependence with the action variable.

**Proposition 2.** *Under the assumption that the graphical representation, corresponding to the environment model, is Markov and faithful to the measured data, $s_t^a \subseteq \mathbf{s_t}$ is a minimal subset of state dimensions that are sufficient for direct control, and $s_{i,t} \in s_t^a$ if and only if $s_{i,t} \not\perp\!\!\!\perp a_{t-1} | \mathbf{s_{t-1}}$.*

Furthermore, based on Proposition 1 and Proposition 2, we can further differentiate $s_t^{ar}, s_t^{\bar{a}r}, s_t^{a\bar{r}}$ from $s_t^r$ and $s_t^a$, which is given in the following proposition.

**Proposition 3.** *Under the assumption that the graphical representation, corresponding to the environment model, is Markov and faithful to the measured data, we can build a connection between the graph structure and statistical independence of causal variables in the RL system, with (1) $s_{i,t} \in s_t^{ar}$ if and only if $s_{i,t} \not\perp\!\!\!\perp R_t | a_{t-1:t}, s_{t-1}^r$ and $s_{i,t} \not\perp\!\!\!\perp a_{t-1} | \mathbf{s_{t-1}}$, (2) $s_{i,t} \in s_t^{\bar{a}r}$ if and only if $s_{i,t} \not\perp\!\!\!\perp R_t | a_{t-1:t}, s_{t-1}^r$ and $s_{i,t} \perp\!\!\!\perp a_{t-1} | \mathbf{s_{t-1}}$, (3) $s_{i,t} \in s_t^{a\bar{r}}$ if and only if $s_{i,t} \perp\!\!\!\perp R_t | a_{t-1:t}, s_{t-1}^r$ and $s_{i,t} \not\perp\!\!\!\perp a_{t-1} | \mathbf{s_{t-1}}$, and (4) $s_{i,t} \in s_t^{\bar{a}\bar{r}}$ if and only if $s_{i,t} \perp\!\!\!\perp R_t | a_{t-1:t}, s_{t-1}^r$ and $s_{i,t} \perp\!\!\!\perp a_{t-1} | \mathbf{s_{t-1}}$.*

By identifying these four categories of latent state variables, we can achieve interpretable input for policies, including (i) reward-relevant and directly controllable parts, (ii) reward-relevant but not controllable parts, (iii) controllable but non-reward-relevant parts, and (iv) unrelated noise. Furthermore, policy training will become more sample efficient and robust to task-irrelevant changes by utilizing only $s_t^{ar}$ and $s_t^{\bar{a}r}$ for policy learning. We show the block-wise identifiability of the four categories of variables in the next section.

# 3 Identifiability Theory

Identifying causally-related latent variables from observations is particularly challenging, as latent variables are generally not uniquely recoverable [16, 17]. Previous work in causal representation learning and nonlinear-ICA has established identifiability conditions for non-parametric temporally latent causal processes [18, 19]. However, these conditions are often too stringent in reality, and none of these methods have been applied to complex control tasks. In contrast to component-wise identifiability, our focus lies on the block-wise identifiability of the four categories of latent state variables. From a policy optimization standpoint, this is sufficient because we do not need the latent variables to be recovered up to permutation and component-wise invertible nonlinearities. This relaxation makes the conditions more likely to hold true and applicable to a wide range of RL tasks. We provide proof of the block-wise identifiability of the four types of latent variables. To the best of our knowledge, we are the first to prove the identifiability of disentangled latent state variables in general nonlinear cases for RL tasks.

According to the causal process in the RL system (as described in Eq.1 in [13]), we can build the following mapping from latent state variables $\mathbf{s}_t$ to observed variables $o_t$ and future cumulative reward $R_t$:

$$[o_t, R_t] = f(s_t^r, s_t^{\bar{r}}, \eta_t), \tag{1}$$

where

$$\begin{array}{rcl} o_t & = & f_1(s_t^r, s_t^{\bar{r}}), \\ R_t & = & f_2(s_t^r, \eta_t). \end{array} \tag{2}$$

Here, note that to recover $s_t^r$, it is essential to take into account all future rewards $r_{t:T}$, because any state dimension $s_{i,t} \in \mathbf{s}_t$ that has a directed path to the future reward $r_{t+\tau}$, for $\tau > 0$, is involved in $s_t^r$. Hence, we consider the mapping from $s_t^r$ to the future cumulative reward $R_t$, and $\eta_t$ represents residuals, except $s_t^r$, that have an effect to $R_t$.

Below, we first provide the definition of blockwise identifiability and relevant notations, related to [20, 21, 22, 23].

**Definition 1** (Blockwise Identifiability). *A latent variable $\mathbf{s_t}$ is blockwise identifiable if there exists a one-to-one mapping $h(\cdot)$ between $\mathbf{s_t}$ and the estimated $\hat{\mathbf{s}}_t$, i.e., $\hat{\mathbf{s}}_t = h(\mathbf{s_t})$.*

**Notations.** We denote by $\tilde{\mathbf{s}}_t := (s_t^r, s_t^{\bar{r}}, \eta_t)$ and by $|s|$ the dimension of a variable $s$. We further denote $d_{s_r} := |s_t^r|$, $d_{s_{\bar{r}}} := |s_t^{\bar{r}}|$, $d_{\tilde{s}} := |\tilde{\mathbf{s}}_t|$, $d_o := |o_t|$, and $d_R := |R_t|$. We denote by $\mathcal{F}$ the support of Jacobian $\mathbf{J}_f(\mathbf{s}_t)$, by $\hat{\mathcal{F}}$ the support of $\mathbf{J}_{\hat{f}}(\hat{\mathbf{s}}_t)$, and by $\mathcal{T}$ the support of $\mathbf{T}(\mathbf{s}_t)$ with $\mathbf{J}_{\hat{f}}(\hat{\mathbf{s}}_t) = \mathbf{J}_f(\mathbf{s}_t)\mathbf{T}(\mathbf{s})$. We also denote $T$ as a matrix with the same support as $\mathcal{T}$.

In addition, given a subset $\mathcal{D} \subseteq \{1, \cdots, d_{\tilde{s}}\}$, the subspace $\mathbb{R}_{\mathcal{D}}^{d_{\tilde{s}}}$ is defined as $\mathbb{R}_{\mathcal{D}}^{d_{\tilde{s}}} := \{z \in \mathbb{R}^{d_{\tilde{s}}} | i \notin \mathcal{D} \implies z_i = 0\}$. In other words, $\mathbb{R}_{\mathcal{D}}^{d_{\tilde{s}}}$ refers to the subspace of $\mathbb{R}^{d_{\tilde{s}}}$ indicated by an index set $\mathcal{D}$.

We next show that the different types of states $s_t^{ar}$, $s_t^{\bar{a}r}$, $s_t^{a\bar{r}}$, and $s_t^{\bar{a}\bar{r}}$ are blockwise identifiable from observed image variable $o_t$, reward variable $r_t$, and action variable $a_t$, under reasonable and weak assumptions, which is partly inspired by [20, 21, 22, 23].

**Theorem 1.** *Suppose that the causal process in the RL system and the four categories of latent state variables can be described as that in Section 2 and illustrated in Figure 1(c). Under the following assumptions*

    *A1. The mapping $f$ in Eq. 1 is smooth and invertible with smooth inverse.*

    *A2. For all $i \in \{1, \ldots, d_o + d_R\}$ and $j \in \mathcal{F}_{i,:}$, there exist $\{\tilde{\mathbf{s}}_t^{(l)}\}_{l=1}^{|\mathcal{F}_{i,:}|}$, so that $span\{\mathbf{J}_f(\tilde{\mathbf{s}}_t^{(l)})_{i,:}\}_{l=1}^{|\mathcal{F}_{i,:}|} = \mathbb{R}_{\mathcal{F}_{i,:}}^{d_{\tilde{s}}}$, and there exists a matrix $T$ with its support identical to that of $\mathbf{J}_{\hat{f}}^{-1}(\hat{\mathbf{s}}_t)\mathbf{J}_f(\tilde{\mathbf{s}}_t)$, so that $\left[\mathbf{J}_f(\tilde{\mathbf{s}}_t^{(l)})T\right]_{j,:} \in \mathbb{R}_{\hat{\mathcal{F}}_{i,:}}^{d_{\tilde{s}}}$.*

*Then, reward-relevant and controllable states $s_t^{ar}$, reward-relevant but not controllable states $s_t^{\bar{a}r}$, reward-irrelevant but controllable states $s_t^{a\bar{r}}$, and noise $s_t^{\bar{a}\bar{r}}$, are blockwise identifiable.*

In the theorem presented above, Assumption $A1$ only assumes the invertibility of function $f$, while functions $f_1$ and $f_2$ are considered general and not necessarily invertible, as that in [23]. Since the function $f$ is the mapping from all (latent) variables, including noise factors, that influence the observed variables, the invertibility assumption holds reasonably. However, note that it is not

reasonable to assume the invertibility of the function $f_2$ since usually, the reward function is not invertible. Intuitively, Assumption *A2* requires that the Jacobian varies "enough" so that it cannot be contained in a proper subspace of $\mathbb{R}^{d_{\bar{s}}}_{\mathcal{F}_{i,:}}$. This requirement is necessary to avoid undesirable situations where the problem becomes ill-posed and is essential for identifiability. A special case when this property does not hold is when the function $f$ is linear, as the Jacobian remains constant in such cases. Some proof techniques of Theorem 1 follow from [20, 22, 23], with the detailed proof given in Appendix A.4.

# 4 World Model with Disentangled Latent Dynamics

Based on the four categories of latent state variables, we formulate a world model with disentangled latent dynamics. Each component of the world model is described as follows:

$$
\begin{cases}
\textit{Observation Model:} & p_\theta\left(o_t \mid \mathbf{s_t}\right) \\
\textit{Reward Model:} & p_\theta\left(r_t \mid s_t^r\right) \\
\textit{Transition Model:} & p_\gamma\left(\mathbf{s_t} \mid \mathbf{s_{t-1}}, a_{t-1}\right) \\
\textit{Representation Model:} & q_\phi\left(\mathbf{s_t} \mid o_t, \mathbf{s_{t-1}}, a_{t-1}\right)
\end{cases}
\tag{3}
$$

Specifically, the world model includes three generative models: an observation model, a reward function, and a transition model (prior), and a representation model (posterior). The observation model and reward model are parameterized by $\theta$. The transition model is parameterized by $\gamma$ and the representation model is parameterized by $\phi$. We assume noises are i.i.d for all models. The action $a_{t-1}$ has a direct effect on the latent states $\mathbf{s_t}$, but not on the perceived signals $o_t$. The perceived signals, $o_t$, are generated from the underlying states $\mathbf{s_t}$, while signals $r_t$ are generated only from reward-relevant latent variables $s_t^r$.

According to the graphical model depicted in Figure 1(c), the transition model and the representation model can be further decomposed into four distinct sub-models, according to the four categories of latent state variables, as shown in equation 4.

$$
\begin{array}{ll}
\textit{Disentangled Transition Model:} & \textit{Disentangled Representation Model:} \\
\begin{cases}
p_{\gamma_1}(s_t^{ar} \mid s_{t-1}^r, a_{t-1}) \\
p_{\gamma_2}(s_t^{\bar{a}r} \mid s_{t-1}^r) \\
p_{\gamma_3}(s_t^{a\bar{r}} \mid \mathbf{s_{t-1}}, a_{t-1}) \\
p_{\gamma_4}(s_t^{\bar{a}\bar{r}} \mid \mathbf{s_{t-1}})
\end{cases}
&
\begin{cases}
q_{\phi_1}(s_t^{ar} \mid o_t, s_{t-1}^r, a_{t-1}) \\
q_{\phi_2}(s_t^{\bar{a}r} \mid o_t, s_{t-1}^r) \\
q_{\phi_3}(s_t^{a\bar{r}} \mid o_t, \mathbf{s_{t-1}}, a_{t-1}) \\
q_{\phi_4}(s_t^{\bar{a}\bar{r}} \mid o_t, \mathbf{s_{t-1}})
\end{cases}
\end{array}
\tag{4}
$$

Specifically, we have $p_\gamma = p_{\gamma_1} \cdot p_{\gamma_2} \cdot p_{\gamma_3} \cdot p_{\gamma_4}$ and $q_\phi = q_{\phi_1} \cdot q_{\phi_2} \cdot q_{\phi_3} \cdot q_{\phi_4}$. Note this factorization differs from previous works [11, 9] that assume independent latent processes. Instead, we only assume conditional independence among the four categories of latent variables given $\mathbf{s_{t-1}}$, which provides a more general factorization. In particular, the dynamics of $s_t^{ar}$ and $s_t^{\bar{a}r}$ are dependent on $s_{t-1}^r$, ensuring that they are unaffected by any reward-irrelevant latent variables present in $s_{t-1}^{\bar{r}}$. On the other hand, $s_t^{\bar{r}}$ may be influenced by all the latent variables from the previous time step. Note that the connections between $\mathbf{s_{t-1}}$ and $s_t^{\bar{r}}$ can be adjusted based on the concrete problem. Since $s_t^{ar}$ and $s_t^{a\bar{r}}$ are controllable variables, their determination also relies on $a_{t-1}$.

## 4.1 World Model Estimation

The optimization of the world model involves joint optimization of its four components to maximize the variational lower bound [24] or, more generally, the variational information bottleneck [25, 26]. The bound encompasses reconstruction terms for both observations and rewards, along with a KL regularizer:

$$
\mathcal{J}_O^t = \ln p_\theta\left(o_t \mid \mathbf{s_t}\right) \quad \mathcal{J}_R^t = \ln p_\theta\left(r_t \mid s_t^r\right) \quad \mathcal{J}_D^t = -\text{KL}\left(q_\phi \| p_\gamma\right).
\tag{5}
$$

Further, the KL regularizer $\mathcal{J}_D^t$ can be decomposed into four components based on our factorization of the state variables. We introduce additional hyperparameters to regulate the amount of information contained within each category of variables:

$$
\mathcal{J}_D^t = -\beta_1 \cdot \text{KL}\left(q_{\phi_1} \| p_{\gamma_1}\right) - \beta_2 \cdot \text{KL}\left(q_{\phi_2} \| p_{\gamma_2}\right) - \beta_3 \cdot \text{KL}\left(q_{\phi_3} \| p_{\gamma_3}\right) - \beta_4 \cdot \text{KL}\left(q_{\phi_4} \| p_{\gamma_4}\right).
\tag{6}
$$

Additionally, we introduce two supplementary objectives to explicitly capture the distinctive characteristics of the four distinct representation categories. Specifically, we characterize the reward-

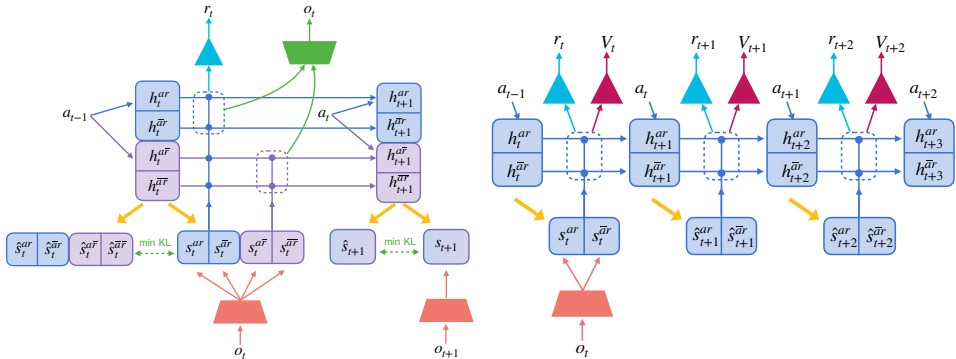

| (a) Learning world model from experiences. | (b) Learning policy by imagination. |

Figure 2: (a) The agent learns the disentangled latent dynamics from prior experiences. The yellow arrow represents a one-one mapping from $h_t^*$ to $s_t^*$ with the same superscript. (b) Within the latent space, state values and actions are forecasted to maximize future value predictions by backpropagating gradients through imagined trajectories. Only $s_t^r$ (reward-relevant) are used for policy optimization.

relevant representations by measuring the dependence between $s_t^r$ and $R_t$, given $a_{t-1:t}$ and $s_{t-1}^r$, that is $I(s_t^r, R_t \mid a_{t-1:t}, s_{t-1}^r)$. To ensure that $s_t^r$ are minimally sufficient for policy training, we maximize $I(s_t^r, R_t \mid a_{t-1:t}, s_{t-1}^r)$ while minimizing $I(s_t^{\bar{r}}, R_t \mid a_{t-1:t}, s_{t-1}^r)$ to discourage the inclusion of redundant information in $s_t^{\bar{r}}$ concerning the rewards:

$$I(s_t^r; R_t \mid a_{t-1:t}, s_{t-1}^r) - I(s_t^{\bar{r}}; R_t \mid a_{t-1:t}, s_{t-1}^r). \tag{7}$$

The conditional mutual information can be expressed as the disparity between two mutual information.

$$
\begin{aligned}
I(s_t^r; R_t \mid a_{t-1:t}, s_{t-1}^r) &= I(R_t; s_t^r, a_{t-1:t}, s_{t-1}^r) - I(R_t; a_{t-1:t}, s_{t-1}^r), \\
I(s_t^{\bar{r}}; R_t \mid a_{t-1:t}, s_{t-1}^r) &= I(R_t; s_t^{\bar{r}}, a_{t-1:t}, s_{t-1}^r) - I(R_t; a_{t-1:t}, s_{t-1}^r).
\end{aligned}
\tag{8}
$$

After removing the common term, we leverage mutual information neural estimation [27] to approximate the value of mutual information. Thus, we reframe the objective in Eq.7 as follows:

$$\mathcal{J}_{\text{RS}}^t = \lambda_1 \cdot \{ I_{\alpha_1}(R_t; s_t^r, a_{t-1:t}, \mathbf{sg}(s_{t-1}^r)) - I_{\alpha_2}(R_t; s_t^{\bar{r}}, a_{t-1:t}, \mathbf{sg}(s_{t-1}^r)) \}. \tag{9}$$

We employ additional neural networks, parameterized by $\alpha$, to estimate the mutual information. To incorporate the conditions from the original objective, we apply the stop_gradient operation $\mathbf{sg}$ to the variable $s_{t-1}^r$. Similarly, to ensure that the representations $s_t^a$ are directly controllable by actions, while $s_t^{\bar{a}}$ are not, we maximize the following objective:

$$\mathcal{J}_{\text{AS}}^t = \lambda_2 \cdot \{ I_{\alpha_3}(a_{t-1}; s_t^a, \mathbf{sg}(\mathbf{s_{t-1}})) - I_{\alpha_4}(a_{t-1}; s_t^{\bar{a}}, \mathbf{sg}(\mathbf{s_{t-1}})) \}. \tag{10}$$

Intuitively, these two objective functions ensure that $s_t^r$ is predictive of the reward, while $s_t^{\bar{r}}$ is not; similarly, $s_t^a$ can be predicted by the action, whereas $s_t^{\bar{a}}$ cannot. The total objective for learning the world model is summarized as:

$$\mathcal{J}_{\text{TOTAL}} = \mathbb{E}_{q_\phi} \left( \sum_t \left( \mathcal{J}_{\text{O}}^t + \mathcal{J}_{\text{R}}^t + \mathcal{J}_{\text{D}}^t + \mathcal{J}_{\text{RS}}^t + \mathcal{J}_{\text{AS}}^t \right) \right) + \text{ const }. \tag{11}$$

The expectation is computed over the dataset and the representation model. Throughout the model learning process, the objectives for estimating mutual information and learning the world model are alternately optimized. A thorough derivation and discussion of the objective function are given in Appendix B.

It is important to note that our approach to learning world models with identifiable factorization is independent of the specific policy optimization algorithm employed. In this work, we choose to build upon the state-of-the-art method Dreamer [6, 7], which iteratively performs exploration, model-fitting, and policy optimization. As illustrated in Figure 2, the learning process for the dynamics involves the joint training of the four categories of latent variables. However, only the variables of $s_t^r$ are utilized for policy learning. We provide the pseudocode for the IFactor algorithm in Appendix C.

# 5 Experiments

In this section, we begin by evaluating the identifiability of our method using simulated datasets. Subsequently, we visualize the learned representations of the four categories in a cartpole environment that includes distractors. Further, we assess the advantages of our factored world model in policy learning by conducting experiments on variants of Robodesk and DeepMind Control Suite. The reported results are aggregated over 5 runs for the Robodesk experiments and over 3 runs for others. A detailed introduction to each environment is provided in Appendix D, while experiment details and hyperparameters can be found in Appendix E. The source code is available at https://github.com/AlexLiuyuren/IFactor

## 5.1 Latent State Identification Evaluation

### 5.1.1 Synthetic environments

**Evaluation Metrics and Baselines**  We generate synthetic datasets that satisfy the identifiability conditions outlined in the theorems (see Appendix D.1). To evaluate the block-wise identifiability of the four categories of representations, we follow the experimental methodology of [28] and compute the coefficient of determination $R^2$ from $s_t^*$ to $\hat{s}_t^*$, as well as from $\hat{s}_t^*$ to $s_t^*$, for $* \in \{ar, \bar{a}r, a\bar{r}, \bar{a}\bar{r}\}$. $R^2$ can be viewed as the identifiability score. $R^2 = 1$ in both directions suggests a one-to-one mapping between the true latent variables and the recovered ones. We compare our latent representation learning method with Denoised MDP [9]. We also include baselines along the line of nonlinear ICA: BetaVAE [29] and FactorVAE [30] which do not consider temporal dependencies; SlowVAE [31] and PCL[32] which leverage temporal constraints but assume independent sources; and TDRL [19] which incorporates temporal constraints and non-stationary noise but does not utilize the action and reward information in the Markov Decision Process.

**Results**  Figure 9 demonstrates the effectiveness of our method in accurately recovering the four categories of latent state variables, as evidenced by high $R^2$ values ($> 0.9$). In contrast, the baseline methods exhibit distortions in the identification results. Particularly, Denoised MDP assumes independent latent process for $x_t, y_t$, and the existence of instantaneous causal effect from $x_t, y_t$ to $z_t$ (see Figure 1(d)), leading to unidentifiable latent variables without further intervention. BetaVAE, FactorVAE, SlowVAE, and PCL, which assume independent sources, show subpar performance. Although TDRL allows for causally-related processes under conditional independence assumptions, it fails to explicitly leverage the distinct transition structures of the four variable categories. More results on the identifiability scores of baselines are provided in Appendix E.1.

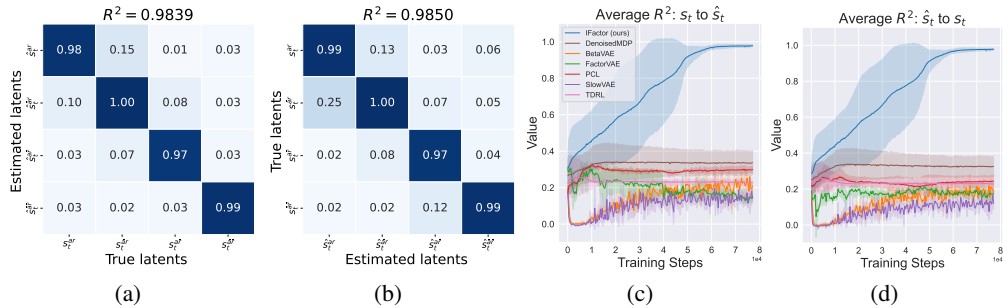

Figure 3: Simulation results: (a) The coefficient of determination ($R^2$) obtained by using kernel ridge regression to regress estimated latents on true latents.(b) The $R^2$ obtained by using kernel ridge regression [33] to regress true latents on estimated latents. (c) The average $R^2$ over four types of representations during training (True latents → Estimated latents).(d) The average $R^2$ over four types of representations during training (Estimated latents → True latents).

### 5.1.2 Modified Cartpole with distractors

We have introduced a variant of the original Cartpole environment by incorporating two distractors. The first distractor is an uncontrollable Cartpole located in the upper portion of the image, which is irrelevant to the rewards. The second distractor is a controllable but reward-irrelevant green light positioned below the reward-relevant Cartpole in the lower part of the image.

After estimating the representation model, we utilize latent traversal to visualize the four categories of representations in the modified Cartpole environment (see Figure 4). Specifically, the recovered $s_t^{ar}$ corresponds to the position of the cart, while $s_t^{\bar{a}r}$ corresponds to the angle of the pole. This demonstrates the ability of our method to automatically factorize reward-relevant representations based on (one-step) controllability: the cart experiences a force at the current time step, whereas the pole angle is influenced in the subsequent time step. Note that during the latent traversal of $s_t^{\bar{a}r}$, the position of the cart remains fixed, even though the pole originally moves with the cart in the video. Additionally, $s_t^{a\bar{r}}$ corresponds to greenness of the light, and $s_t^{\bar{a}\bar{r}}$ corresponds to the uncontrollable and reward-irrelevant cartpole. These findings highlight the capability of our representation learning method to disentangle and accurately recover the ground-true latent variables from videos. The identifiability scores of four categories of representations are provided in Appendix E.2.

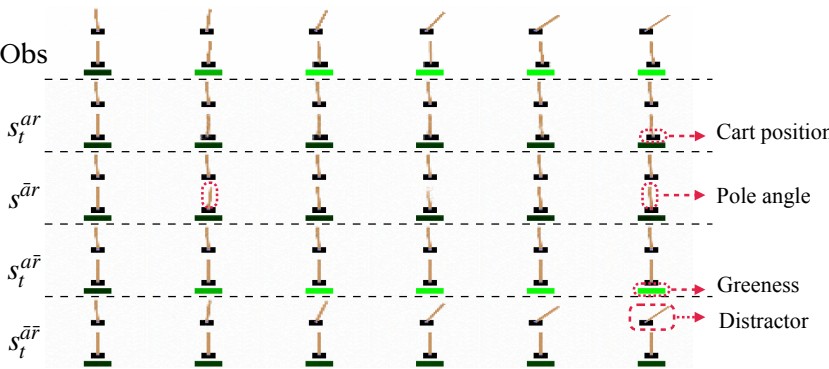

Figure 4: Latent traversal for four types of representations in the modified Cartpole environment.

## 5.2 Policy Optimization Evaluation

To assess the effectiveness of our method in enhancing policy learning by utilizing minimal yet sufficient representations for control, we conducted experiments on more complex control tasks. One of these tasks is a variant of Robodesk [34], which includes realistic noise elements such as flickering lights, shaky cameras, and a dynamic video background. In this task, the objective for the agent is to change the hue of a TV screen to green using a button press. We also consider variants of DeepMind Control Suite (DMC) [35, 9], where distractors such as a dynamic video background, noisy sensor readings, and a jittering camera are introduced to the original DMC environment. These additional elements aim to challenge the agent's ability to focus on the relevant aspects of the task while filtering out irrelevant distractions. Baseline results except for DreamerPro [36] are derived from the Denoised MDP paper. We have omitted the standard deviation of their performance for clarity.

**Evaluation Metrics and Baselines**  We evaluate the performance of the policy at intervals of every 10,000 environment steps and compare our method with both model-based and model-free approaches. Among the model-based methods, we include Denoised MDP [9], which is the state-of-the-art method for variants of Robodesk and DMC. We also include DreamerPro [36], TIA [11] and Dreamer [6] as additional model-based baselines. For the model-free methods, we include DBC [37], CURL[38], and PI-SAC [39]. To ensure a fair comparison, we have aligned all common hyperparameters and neural network structures with those used in the Denoised MDP [9].

### 5.2.1 RoboDesk with Various Noise Distractors

In Figure 5, the left image demonstrates that our model IFactor achieves comparable performance to Denoised MDP while outperforming other baselines. The results of baselines except for DreamerPro are directly copied from the paper of Denoised MDP [9] (its error bar cannot be directly copied), and the replication of Denoised MDP results is given in Appendix E.3. Furthermore, to investigate whether $s_t^r$ serves as minimal and sufficient representations for policy learning, we retrain policies using the Soft Actor-Critic algorithm [40] with different combinations of the four learned categories as input. Remarkably, the policy trained using $s_t^r$ exhibits the best performance, while the performance of other policies degrades due to insufficient information (e.g., $s_t^{ar}$ and $s_t^{\bar{r}}$) or redundant information

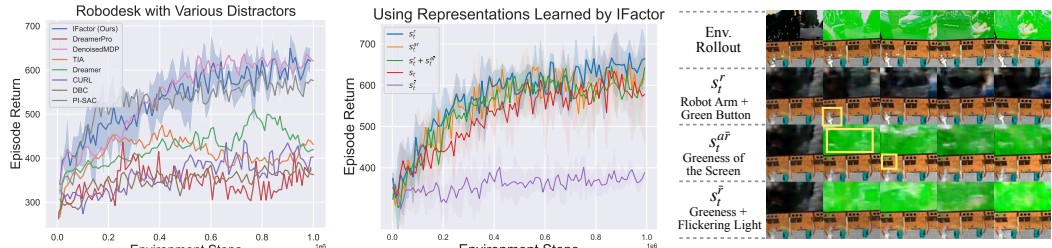

Figure 5: Results on Robodesk. We have omitted the standard deviation of the performance of baselines for clarity. The left image shows that our method IFactor achieves comparable performance with Denoised Mdp, outperforming other baselines. The middle image shows the policy optimization process by SAC [40] using representations learned by IFactor, where policies that use $s_t^r$ as input achieve the best performance. The right image shows the latent traversal of the representations.

(e.g., $s_t^r + s_t^{a\bar{r}}$ and $\mathbf{s_t}$). Moreover, we conduct latent traversal on the learned representations to elucidate their original meaning in the observation. Interestingly, our model identifies the greenness of the screen as $s_t^{a\bar{r}}$, which may initially appear counter-intuitive. However, this phenomenon arises from the robot arm pressing the green button, resulting in the screen turning green. Consequently, the screen turning green becomes independent of rewards conditioned on the robot arm pressing the green button, aligning with the definition of $s_t^{a\bar{r}}$. This empirically confirms the minimality of $s_t^r$ for policy optimization.

### 5.2.2 DeepMind Control Suite (DMC)

Figure 6 demonstrates the consistent superiority of our method over baselines in the Deepmind Control Suite (DMC) across various distractor scenarios. Our approach exhibits robust performance in both noiseless and noisy environments, demonstrating the effectiveness of our factored world model in eliminating distractors while preserving essential information for effective and efficient control in policy learning. The ablation study of IFactor's objective function terms is presented in Appendix B.1 and Appendix E.3, which shows that the inclusion of mutual information constraints is essential to promote disentanglement and enhance policy performance. The policy performance that includes the standard error is provided in Appendix E.4. Visualization of the learned representations is also provided in Appendix E.5.

## 6 Related Work

**World Model Learning in RL.** Image reconstruction [41, 42] and contrastive learning [43, 44, 45, 46, 47] are wildly used to learn representations in RL. Dreamer and its subsequent extensions [5, 6, 7, 8] adopt a world model-based learning approach, where the policy is learned solely through dreaming in the world model. However, the agents trained by these techniques tend to underperform in noisy environments [9]. To address this issue, numerous approaches have been introduced to enhance robustness to distractions. Task Informed Abstractions (TIA) [11] explicitly partition the latent state space into reward-relevant and reward-irrelevant features. Denoised MDP [9] takes a step further and decomposes the reward-relevant states into controllable and uncontrollable components. InfoPower [48] prioritizes information that is correlated with action based on mutual information. Iso-Dream [12] learns controllable and noncontrollable sources of spatiotemporal changes on isolated state transition branches. Our method extends these work by accommodating a more general factorization with block-wise identifiablity. Recent research also explores reconstruction-free representation learning methods [36, 49, 50, 51]. DreamerPro [36] combines Dreamer with prototypes, which distills temporal structures from past observations and actions. Temporal Predictive Coding [50] encodes elements in the environment that can be predicted across time. Contrastively-trained Structured World Models [51] utilize graph neural networks and a contrastive approach for representation learning in environments with compositional structure. Latent states learned by these methods are not identifiable due to the reconstruction-free property, where the mixing function is not invertible anymore. A detailed comparison between IFactor and related work is also given in Appendix F.

**Identifiability in Causal Representation Learning.** Temporal structure and nonstationarities were recently used to achieve identifiability in causal representation learning [52]. Methods such as TCL

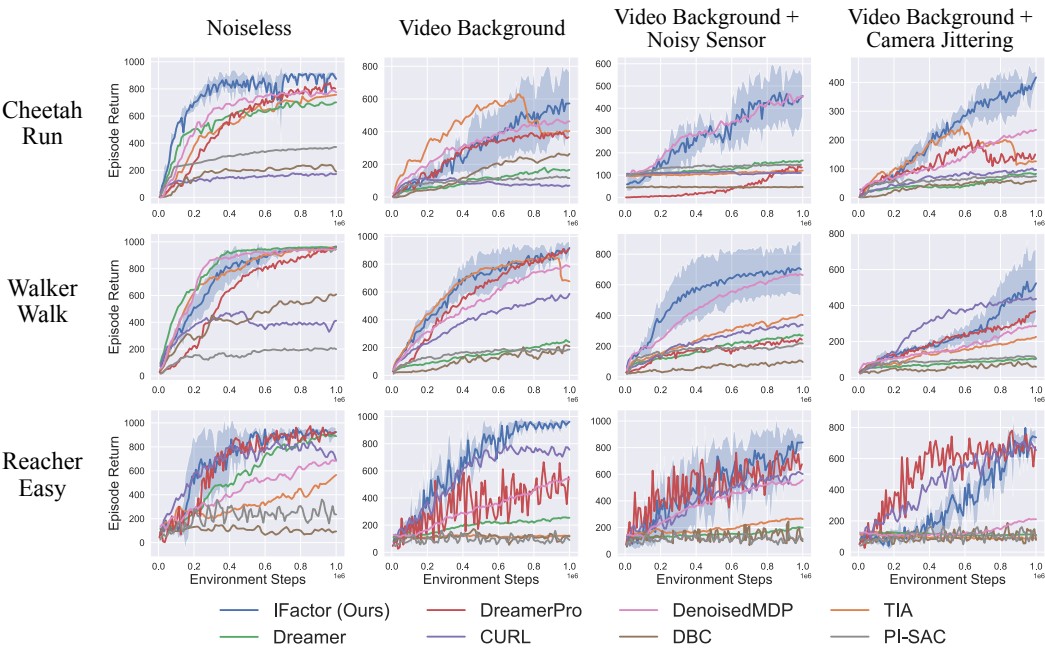

Figure 6: Policy optimization on variants of DMC with various distractors.

[53] and PCL [32] leverage nonstationarity in temporal data through contrastive learning to identify independent sources. CITRIS [54] and iCITRIS [55] takes advantage of observed intervention targets to identify causal factors. LEAP [18] introduces identifiability conditions for both parametric and nonparametric latent processes. TDRL [19] explores identifiability of latent processes under stationary environments and distribution shifts. ASRs [13] establish the identifiability of the representations in the linear-gaussian case in the RL setting. Our work extends the theoretical results in ASRs to enable block-wise identifiability of four categories of latent variables in general nonlinear cases.

## 7 Conclusion

In this paper, we present a general framework to model four distinct categories of latent state variables within the RL system, based on their interactions with actions and rewards. We establish the block-wise identifiability of these latent categories in general nonlinear cases, under weak and realistic assumptions. Accordingly, we propose IFactor to extract four types of representations from raw observations and use reward-relevant representations for policy optimization. Experiments verify our theoretical results and show that our method achieves state-of-the-art performance in variants of the DeepMind Control Suite and RoboDesk. The basic limitation of this work is that the underlying latent processes are assumed to have no instantaneous causal relations but only time-delayed influences. This assumption does not hold true if the resolution of the time series is much lower than the causal frequency. We leave the identifiability of the latent dynamics in the presence of instantaneous causal and the extension of our method to heterogeneous environments to future work.

## 8 Acknowledgements

This work is supported by National Key Research and Development Program of China (2020AAA0107200), the Major Key Project of PCL (PCL2021A12), China Scholarship Council, NSF Grant 2229881, the National Institutes of Health (NIH) under Contract R01HL159805, a grant from Apple Inc., a grant from KDDI Research Inc., and generous gifts from Salesforce Inc., Microsoft Research, and Amazon Research. We thank Guangyi Chen, Weiran Yao and the anonymous reviewers for their support and helpful discussions on improving the paper.

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

# A  Theoretical Proofs

## A.1  Proof of Proposition 1

Proposition 1 shows that $s_t^r$, which has directed paths to $r_{t+\tau}$ (for $\tau \geq 0$), is minimally sufficient for policy learning that aims to maximize the future reward and can be characterized by conditional dependence with the cumulative reward variable $R_t$.

**Proposition 1.** *Under the assumption that the graphical representation, corresponding to the environment model, is Markov and faithful to the measured data, $s_t^r \subseteq \mathbf{s_t}$ is a minimal subset of state dimensions that are sufficient for policy learning, and $s_{i,t} \in s_t^r$ if and only if $s_{i,t} \not\!\perp\!\!\!\perp R_t | a_{t-1:t}, s_{t-1}^r$.*

We first give the definitions of the Markov condition and the faithfulness assumption [56, 57], which will be used in the proof.

**Definition 1** (Global Markov Condition). *The distribution $p$ over a set of variables $\mathbf{V}$ satisfies the global Markov property on graph $G$ if for any partition $(A, B, C)$ such that if $B$ d-separates $A$ from $C$, then $p(A, C|B) = p(A|B)p(C|B)$.*

**Definition 2** (Faithfulness Assumption). *There are no independencies between variables that are not entailed by the Markov Condition in the graph.*

Below, we give the proof of Proposition 1.

*Proof.* The proof contains the following three steps.

- In step 1, we show that a state dimension $s_{i,t}$ is in $s_t^r$, that is, it has a directed path to $r_{t+\tau}$, if and only if $s_{i,t} \not\!\perp\!\!\!\perp R_t | a_{t-1:t}, \mathbf{s}_{t-1}$.

- In step 2, we show that for $s_{i,t}$ with $s_{i,t} \not\!\perp\!\!\!\perp R_t | a_{t-1:t}, \mathbf{s}_{t-1}$, if and only if $s_{i,t} \not\!\perp\!\!\!\perp R_t | a_{t-1:t}, s_{t-1}^r$.

- In step 3, we show that $s_t^r$ are minimally sufficient for policy learning.

**Step 1:**  We first show that if a state dimension $s_{i,t}$ is in $s_t^r$, then $s_{i,t} \not\!\perp\!\!\!\perp R_t | a_{t-1:t}, \mathbf{s}_{t-1}$.

We prove it by contradiction. Suppose that $s_{i,t}$ is independent of $R_t$ given $a_{t-1:t}$ and $\mathbf{s}_{t-1}$. Then according to the faithfulness assumption, we can see from the graph that $s_{i,t}$ does not have a directed path to $r_{t+\tau}$, which contradicts the assumption, because, otherwise, $a_{t-1:t}$ and $\mathbf{s}_{t-1}$ cannot break the paths between $s_{i,t}$ and $R_t$ which leads to the dependence.

We next show that if $s_{i,t} \not\!\perp\!\!\!\perp R_t | a_{t-1:t}, \mathbf{s}_{t-1}$, then $s_{i,t} \in s_t^r$.

Similarly, by contradiction suppose that $s_{i,t}$ does not have a directed path to $r_{t+\tau}$. From the graph, it is easy to see that $a_{t-1:t}$ and $\mathbf{s}_{t-1}$ must d-separate the path between $s_{i,t}$ and $R_t$. According to the Markov assumption, $s_{i,t}$ is independent of $R_t$ given $a_{t-1:t}$ and $\mathbf{s}_{t-1}$, which contradicts to the assumption. Since we have a contradiction, it must be that $s_{i,t}$ has a directed path to $r_{t+\tau}$, i.e. $s_{i,t} \in s_t^r$.

**Step 2:**  In step 1, we have shown that $s_{i,t} \not\!\perp\!\!\!\perp R_t | a_{t-1:t}, \mathbf{s}_{t-1}$, if and only if it has a directed path to $r_{t+\tau}$. From the graph, it is easy to see that for those state dimensions which have a directed path to $r_{t+\tau}$, $a_{t-1:t}$ and $\mathbf{s}_{t-1}$ cannot break the path between $s_{i,t}$ and $R_t$. Moreover, for those state dimensions which do not have a directed path to $r_{t+\tau}$, $a_{t-1:t}$ and $s_{t-1}^r$ are enough to break the path between $s_{i,t}$ and $R_t$.

Therefore, for $s_{i,t}$, $s_{i,t} \not\!\perp\!\!\!\perp R_t | a_{t-1:t}, \mathbf{s}_{t-1}$, if and only if $s_{i,t} \not\!\perp\!\!\!\perp R_{t+1} | a_{t-1:t}, s_{t-1}^r$.

**Step 3:**  In the previous steps, it has been shown that if a state dimension $s_{i,t}$ is in $s_t^r$, then $s_{i,t} \not\!\perp\!\!\!\perp R_t | a_{t-1:t}, s_{t-1}^r$, and if a state dimension $s_{i,t}$ is not in $s_t^r$, then $s_{i,t} \perp\!\!\!\perp R_t | a_{t-1:t}, s_{t-1}^r$. This implies that $s_t^r$ are minimally sufficient for policy learning to maximize the future reward. □

## A.2  Proof of Proposition 2

Moreover, the proposition below shows that $s_t^a$, which receives an edge from $a_{t-1}$, can be directly controlled by actions and can be characterized by conditional dependence with the action variable.

**Proposition 2.** *Under the assumption that the graphical representation, corresponding to the environment model, is Markov and faithful to the measured data, $s_t^a \subseteq \mathbf{s_t}$ is a minimal subset of state dimensions that are sufficient for direct control, and $s_{i,t} \in s_t^a$ if and only if $s_{i,t} \not\!\perp\!\!\!\perp a_{t-1} | \mathbf{s_{t-1}}$.*

Below, we give the proof of Proposition 2.

*Proof.* The proof contains the following two steps.

- In step 1, we show that a state dimension $s_{i,t}$ is in $s_t^a$, that is, it receives an edge from $a_{t-1}$, if and only if $s_{i,t} \not\perp a_{t-1}|\mathbf{s_{t-1}}$.

- In step 2, we show that $s_t^a$ contains a minimally sufficient subset of state dimensions that can be directly controlled by actions.

**Step 1:** We first show that if a state dimension $s_{i,t}$ is in $s_t^a$, then $s_{i,t} \not\perp a_{t-1}|\mathbf{s_{t-1}}$.

We prove it by contradiction. Suppose that $s_{i,t}$ is independent of $a_{t-1}$ given $\mathbf{s}_{t-1}$. Then according to the faithfulness assumption, we can see from the graph that $s_{i,t}$ does not receive an edge from $a_{t-1}$, which contradicts the assumption, because, otherwise, $\mathbf{s}_{t-1}$ cannot break the paths between $s_{i,t}$ and $a_{t-1}$ which leads to the dependence.

We next show that if $s_{i,t} \not\perp a_{t-1}|\mathbf{s_{t-1}}$, then $s_{i,t} \in s_t^a$.

Similarly, by contradiction suppose that $s_{i,t}$ does not receive an edge from $a_{t-1}$. From the graph, it is easy to see that $\mathbf{s}_{t-1}$ must break the path between $s_{i,t}$ and $a_{t-1}$. According to the Markov assumption, $s_{i,t}$ is independent of $a_{t-1}$ given $\mathbf{s}_{t-1}$, which contradicts to the assumption. Since we have a contradiction, it must be that $s_{i,t}$ has an edge from $a_{t-1}$.

**Step 2:** In the previous steps, it has been shown that if a state dimension $s_{i,t}$ is in $s_t^a$, then $s_{i,t} \not\perp a_{t-1}|\mathbf{s_{t-1}}$, and if a state dimension $s_{i,t}$ is not in $s_t^a$, then $s_{i,t} \perp a_{t-1}|\mathbf{s_{t-1}}$. This implies that $s_t^a$ is minimally sufficient for one-step direct control. □

## A.3 Proof of Proposition 3

Furthermore, based on Proposition 1 and Proposition 2, we can further differentiate $s_t^{ar}, s_t^{\bar{a}r}, s_t^{a\bar{r}}$ from $s_t^r$ and $s_t^a$, which is given in the following proposition.

**Proposition 3.** *Under the assumption that the graphical representation, corresponding to the environment model, is Markov and faithful to the measured data, we can build a connection between the graph structure and statistical independence of causal variables in the RL system, with (1) $s_{i,t} \in s_t^{ar}$ if and only if $s_{i,t} \not\perp R_t|a_{t-1}, s_{t-1}^r$ and $s_{i,t} \not\perp a_{t-1}|\mathbf{s_{t-1}}$, (2) $s_{i,t} \in s_t^{\bar{a}r}$ if and only if $s_{i,t} \not\perp R_t|a_{t-1}, s_{t-1}^r$ and $s_{i,t} \perp a_{t-1}|\mathbf{s_{t-1}}$, (3) $s_{i,t} \in s_t^{a\bar{r}}$ if and only if $s_{i,t} \perp R_t|a_{t-1}, s_{t-1}^r$ and $s_{i,t} \not\perp a_{t-1}|\mathbf{s_{t-1}}$, and (4) $s_{i,t} \in s_t^{\bar{a}\bar{r}}$ if and only if $s_{i,t} \perp R_t|a_{t-1}, s_{t-1}^r$ and $s_{i,t} \perp a_{t-1}|\mathbf{s_{t-1}}$.*

*Proof.* This proposition can be easily proved by levering the results from Propositions 1 and 2. □

## A.4 Proof of Theorem 1

According to the causal process in the RL system (as described in Eq.1 in [13]), we can build the following mapping from latent state variables $\mathbf{s}_t$ to observed variables $o_t$ and future cumulative reward $R_t$:

$$[o_t, R_t] = f(s_t^r, s_t^{\bar{r}}, \eta_t), \tag{12}$$

where

$$\begin{aligned} o_t &= f_1(s_t^r, s_t^{\bar{r}}), \\ R_t &= f_2(s_t^r, \eta_t). \end{aligned} \tag{13}$$

Here, note that to recover $s_t^r$, it is essential to take into account all future rewards $r_{t:T}$, because any state dimension $s_{i,t} \in \mathbf{s}_t$ that has a directed path to the future reward $r_{t+\tau}$, for $\tau > 0$, is involved in $s_t^r$. Hence, we consider the mapping from $s_t^r$ to the future cumulative reward $R_t$, and $\eta_t$ represents residuals, except $s_t^r$, that have an effect to $R_t$.

The following theorem shows that the different types of states $s_t^{ar}, s_t^{\bar{a}r}, s_t^{a\bar{r}}$, and $s_t^{\bar{a}\bar{r}}$ are blockwise identifiable from observed image variable $o_t$, reward variable $r_t$, and action variable $a_t$, under reasonable and weak assumptions.

**Theorem 1.** *Suppose that the causal process in the RL system and the four categories of latent state variables can be described as that in Section 2 and illustrated in Figure 1(c). Under the following assumptions*

*A1. The mapping f in Eq. 12 is smooth and invertible with smooth inverse.*

*A2. For all $i \in \{1, \ldots, d_o + d_R\}$ and $j \in \mathcal{F}_{i,:}$, there exist $\{\tilde{\mathbf{s}}_t^{(l)}\}_{l=1}^{|\mathcal{F}_{i,:}|}$, so that $\text{span}\{\mathbf{J}_f(\tilde{\mathbf{s}}_t^{(l)})_{i,:}\}_{l=1}^{|\mathcal{F}_{i,:}|} = \mathbb{R}_{\mathcal{F}_{i,:}}^{d_{\tilde{s}}}$, and there exists a matrix T with its support identical to that of $\mathbf{J}_{\hat{f}}^{-1}(\hat{\mathbf{s}}_t)\mathbf{J}_f(\tilde{\mathbf{s}}_t)$, so that $[\mathbf{J}_f(\tilde{\mathbf{s}}_t^{(l)})T]_{j,:} \in \mathbb{R}_{\hat{\mathcal{F}}_{i,:}}^{d_{\tilde{s}}}$.*

*Then, reward-relevant and controllable states $s_t^{ar}$, reward-relevant but not controllable states $s_t^{\bar{a}r}$, reward-irrelevant but controllable states $s_t^{a\bar{r}}$, and noise $s_t^{\bar{a}\bar{r}}$, are blockwise identifiable.*

In the theorem presented above, Assumption *A1* only assumes the invertibility of function $f$, while functions $f_1$ and $f_2$ are considered general and not necessarily invertible, as that in [23]. Since the function $f$ is the mapping from all (latent) variables, including noise factors, that influence the observed variables, the invertibility assumption holds reasonably. However, note that it is not reasonable to assume the invertibility of the function $f_2$ since usually, the reward function is not invertible. Assumption *A2*, which is also given in [20, 22, 23], aims to establish a more generic condition that rules out certain sets of parameters to prevent ill-posed conditions. Specifically, it ensures that the Jacobian is not partially constant. This condition is typically satisfied asymptotically, and it is necessary to avoid undesirable situations where the problem becomes ill-posed.

*Proof.* The proof consists of four steps.

1. In step 1, we show that $s_t^a = s_t^{ar} \cup s_t^{a\bar{r}}$ is blockwise identifiable, by using the characterization that the action variable $a_t$ only directly influences $s_t^{ar}$ and $s_t^{a\bar{r}}$.

2. In step 2, we show that $s_t^r = s_t^{ar} \cup s_t^{\bar{a}r}$ is blockwise identifiable, by using the characterization that the future cumulative reward $R_t$ is only influenced by $s_t^{ar}$ and $s_t^{\bar{a}r}$.

3. In step 3, we show that $s_t^{ar}$ is blockwise identifiable, by using the identifiability of $s_t^{ar} \cup s_t^{a\bar{r}}$ and $s_t^{ar} \cup s_t^{\bar{a}r}$.

4. In step 4, we further show the blockwise identifiability of $s_t^{\bar{a}r}$, $s_t^{a\bar{r}}$, and $s_t^{\bar{a}\bar{r}}$.

**Step 1: prove the block identifiability of $s_t^a$.**

For simplicity of notation, below, we omit the subscript $t$.

Let $h := f^{-1} \circ \hat{f}$. We have

$$\hat{\mathbf{s}} = h(\mathbf{s}), \tag{14}$$

where $h = f^{-1} \circ \hat{f}$ is the transformation between the true latent variable and the estimated one, and $\hat{f} : \mathcal{S} \to \mathcal{X}$ denotes the estimated invertible generating function. Note that as both $f^{-1}$ and $\hat{f}$ are smooth and invertible, $h$ and $h^{-1}$ is smooth and invertible.

Since $h(\cdot)$ is smooth over $\mathcal{S}$, its Jocobian can be written as follows:

$$J_{h^{-1}} = \begin{bmatrix} A := \frac{\partial s^{\bar{a}}}{\partial \hat{s}^{\bar{a}}} & B := \frac{\partial s^{\bar{a}}}{\partial \hat{s}^a} \\ C := \frac{\partial s^a}{\partial \hat{s}^{\bar{a}}} & D := \frac{\partial s^a}{\partial \hat{s}^a} \end{bmatrix} \tag{15}$$

The invertibility of $h^{-1}$ implies that $J_{h^{-1}}$ is full rank. Since $s^a$ has changing distributions over the action variable $a$ while $s^{\bar{a}}$ has invariant distributions over different values of $a$, we can derive that $C = 0$. Furthermore, because $J_{h^{-1}}$ is full rank and $C$ is a zero matrix, $D$ must be of full rank, which implies $h_a'^{-1}$ is invertible, where $h_a'^{-1}$ denotes the first derivative of $h_a^{-1}$. Therefore, $s^a$ is blockwise identifiable up to invertible transformations.

**Step 2: prove the blockwise identifiability of $s_t^r$.**

Recall that we have the following mapping:

$$[o_t, R_t] = f(s_t^r, s_t^{\bar{r}}, \eta_t),$$

where

$$\begin{aligned} o_t &= f_1(s_t^r, s_t^{\bar{r}}), \\ R_t &= f_2(s_t^r, \eta_t). \end{aligned}$$

Note that here to recover $s_t^r$, we need to take into account all future rewards, because $s_t^r$ contains all those state dimensions that have a directed path to future rewards $r_{t+1:T}$. $\eta_t$ represents all other factors, except $s_t^r$, that influence $R_t$ at time instance $t$. Further note here we assume the invertibility of $f$, while $f_1$ and $f_2$ are general functions not necessarily invertible.

We denote by $\tilde{\mathbf{s}} = (s^r, s^{\bar{r}}, \eta)$. We further denote by the dimension of $s^r$ by $d_{s^r}$, the dimension of $s^{\bar{r}}$ by $d_{s^{\bar{r}}}$, the dimension of $\tilde{s}$ by $d_{\tilde{s}}$, the dimension of $o$ by $d_o$, and the dimension of $R_t$ by $d_R$.

We denote by $\mathcal{F}$ the support of $\mathbf{J}_f(\mathbf{s})$, by $\hat{\mathcal{F}}$ the support of $\mathbf{J}_{\hat{f}}(\hat{\mathbf{s}})$, and by $\mathcal{T}$ the support of $\mathbf{T}(\mathbf{s})$. We also denote $T$ as a matrix with the same support as $\mathcal{T}$. The proof technique mainly follows [23], as well as the required assumptions, and is also related to [20, 22].

Since $h := \hat{f}^{-1} \circ f$, we have $\hat{f} = f \circ h^{-1}(\tilde{\mathbf{s}})$. By applying the chain rule repeatedly, we have

$$\mathbf{J}_{\hat{f}}(\hat{\mathbf{s}}) = \mathbf{J}_f(\tilde{\mathbf{s}}) \cdot \mathbf{J}_{h^{-1}}(h(\tilde{\mathbf{s}})). \tag{16}$$

With Assumption A2, for any $i \in \{1, \ldots, d_o + d_R\}$, there exists $\{\tilde{\mathbf{s}}^{(l)}\}_{l=1}^{|\mathcal{F}_{i,:}|}$, s.t. $\mathrm{span}(\{\mathbf{J}_f(\tilde{\mathbf{s}}^{(l)})_{i,:}\}_{l=1}^{|\mathcal{F}_{i,:}|}) = \mathrm{R}_{\mathcal{F}_{i,:}}^{d_{\tilde{s}}}$.

Since $\{\mathbf{J}_f(\tilde{\mathbf{s}}^{(l)})_{i,:}\}_{l=1}^{|\mathcal{F}_{i,:}|}$ forms a basis of $\mathrm{R}_{\mathcal{F}_{i,:}}^{d_{\tilde{s}}}$, for any $j_0 \in \mathcal{F}_{i,:}$, we can write canonical basis vector $e_{j_0} \in \mathrm{R}_{\mathcal{F}_{i,:}}^{d_{\tilde{s}}}$ as:

$$e_{j_0} = \sum_{l \in \mathcal{F}_{i,:}} \alpha_l \cdot \mathbf{J}_g(\tilde{\mathbf{s}}^{(l)})_{i,:}, \tag{17}$$

where $\alpha_l \in \mathrm{R}$ is a coefficient.

Then, following Assumption A2, there exists a deterministic matrix $T$ such that

$$T_{j_0,:} = e_{j_0}^\top T = \sum_{l \in \mathcal{F}_{i,:}} \alpha_l \cdot \mathbf{J}_g(\tilde{\mathbf{s}}^{(l)})_{i,:} T \in \mathrm{R}_{\hat{\mathcal{F}}_{i,:}}^{d_{\tilde{s}}}, \tag{18}$$

where $\in$ is due to that each element in the summation belongs to $\mathrm{R}_{\hat{\mathcal{F}}_{i,:}}^{d_{\tilde{s}}}$.

Therefore,

$$\forall j \in \mathcal{F}_{i,:}, T_{j,:} \in \mathrm{R}_{\hat{\mathcal{F}}_{i,:}}^{d_{\tilde{s}}}.$$

Equivalently, we have:

$$\forall (i, j) \in \mathcal{F}, \quad \{i\} \times T_{j,:} \subset \hat{\mathcal{F}}. \tag{19}$$

We would like to show that $\hat{s}^r$ does not depend on $s^{\bar{r}}$ and $\eta$, that is, $T_{i,j} = 0$ for $i \in \{1, \ldots, d_{s^r}\}$ and $j \in \{d_{s^r} + 1, \ldots, d_{\tilde{s}}\}$.

We prove it by contradiction. Suppose that $\hat{s}^r$ had dependence on $s^{\bar{r}}$, that is, $\exists (j_{s^r}, j_{s^{\bar{r}}}) \in \mathcal{T}$ with $j_{s^r} \in \{1, \ldots, d_{s^r}\}$ and $j_{s^{\bar{r}}} \in \{d_{s^r} + 1, \ldots, d_{s^r} + d_{s^{\bar{r}}}\}$.

Hence, there must exist $i_r \in \{d_o + 1, \ldots, d_o + d_R\}$, such that, $(i_r, j_{s^{\bar{r}}}) \in \mathcal{F}$.

It follows from Equation 19 that:

$$\{i_r\} \times \mathcal{T}_{j_{s^{\bar{r}}},:} \in \hat{\mathcal{F}} \implies (i_r, j_{s^{\bar{r}}}) \in \hat{\mathcal{F}}. \tag{20}$$

However, due to the structure of $\hat{f}_2$, $[\mathbf{J}_{\hat{f}_2}]_{i_r, j_{s^{\bar{r}}}} = 0$, which results in a contradiction. Therefore, such $(i_r, j_{s^{\bar{r}}})$ does not exist and $\hat{s}^r$ does not depend on $s^{\bar{r}}$. The same reasoning implies that $\hat{s}^r$ does not dependent on $\eta$. Thus, $\hat{s}_r$ does not depend on $(s^{\bar{r}}, \eta)$. In conclusion, $\hat{s}^r$ does not contain extra information beyond $s^r$.

Similarly, we can show that $(\hat{s}^{\bar{r}}, \hat{\eta})$ does not contain information of $s_r$.

Therefore, there is a one-to-one mapping between $s^r$ and $\hat{s}^r$.

**Step 3: prove the blockwise identifiability of $s_t^{ar}$.**

In Step 1 and Step 2, we have shown that both $s^a$ and $s^r$ are blockwise identifiable. That is,

$$\begin{aligned} \hat{s}^r &= h_r(s^r), \\ \hat{s}^a &= h_a(s^a), \end{aligned} \tag{21}$$

where $h_a$ and $h_r$ are invertible functions.

According to the invariance relation of $s^{ar}$, We have the following relations:

$$\hat{s}^{ar} = h_r(s^r)_{1:d_{s^{ar}}} = h_a(s^a)_{1:d_{s^{ar}}}. \tag{22}$$

It remains to show that both $\tilde{h}_r := h_r(\cdot)_{1:d_{s^{ar}}}$ and $\tilde{h}_a := h_a(\cdot)_{1:d_{s^{ar}}}$ do not depend on $s^{\bar{a}r}$ and $s^{a\bar{r}}$ in their arguments.

We will prove this by contradiction, following the proof technique in [58]. Without loss of generality, we suppose $\exists l \in \{1, \cdots, d_{s^{\bar{a}r}}\}$, $s^{r*} \in \mathcal{S}^r$, s.t., $\frac{\partial \tilde{h}_r}{\partial s_l^{\bar{a}r}}(s^{r*}) \neq 0$. As $h$ is smooth, it has continuous partial derivatives. Thus, $\frac{\partial \tilde{h}_r}{\partial s_l^{\bar{a}r}} \neq 0$ holds true in a neighbourhood of $s^{r*}$, i.e.,

$$\exists \eta > 0, s.t., s_l^{\bar{a}r} \to \tilde{h}_r(s^{ar*}, (s_{-l}^{\bar{a}r*}, s_l^{\bar{a}r*})) \text{ is strictly monotonic on } (s_l^{\bar{a}r*} - \eta, s_l^{\bar{a}r*} + \eta), \tag{23}$$

where $s_{-l}^{\bar{a}r}$ denotes variable $s^{\bar{a}r}$ excluding the dimension $l$.

We further define an auxiliary function $\psi : \mathcal{S}^{ar} \times \mathcal{S}^{\bar{a}r} \times \mathcal{S}^{a\bar{r}} \to \mathrm{R}_{\geq 0}$ as follows:

$$\psi(s^{ar}, s^{\bar{a}r}, s^{a\bar{r}}) := |\tilde{h}_r(s^r) - \tilde{h}_a(s^a)|. \tag{24}$$

To obtain the contradiction to the invariance, it remains to show that $\psi > 0$ with a probability greater than zero w.r.t. the true generating process.

There are two situations at $(s^{ar*}, s^{\bar{a}r*}, s^{a\bar{r}*})$ where $s^{\bar{a}r*}$ is an arbitrary point in $\mathcal{S}^{\bar{a}r}$:

- situation 1: $\psi(s^{ar*}, s^{\bar{a}r*}, s^{a\bar{r}*}) > 0$;

- situation 2: $\psi(s^{ar*}, s^{\bar{a}r*}, s^{a\bar{r}*}) = 0$.

In situation 1, we have identified a specific point $\psi(s^{ar*}, s^{\bar{a}r*}, s^{a\bar{r}*})$ that makes $\psi > 0$.

In situation 2, Eq. 23 implies that $\forall s_l^{\bar{a}r} \in (s_l^{ar*}, s_l^{ar*} + \eta)$, $\psi(s^{ar*}, (s_{-l}^{ar*}, s_l^{\bar{a}r}), s^{a\bar{r}*}) > 0$.

Thus, in both situations, we can locate a point $(s^{ar*}, s^{\bar{a}r*'}, s^{a\bar{r}*})$ such that $\psi(s^{ar*}, s^{\bar{a}r*'}, s^{a\bar{r}*}) > 0$, where $s^{\bar{a}r*'} = s^{\bar{a}r*}$ in situation 1 and $s_l^{ar*'} \in (s_l^{ar*}, s_l^{ar*} + \eta), s_{-l}^{ar*'} = s_{-l}^{ar*}$ in situation 2.

Since $\psi$ is a composition of continuous functions, it is continuous. As pre-image of open sets are always open for continuous functions, the open set $\mathrm{R}_{>0}$ has an open set $\mathcal{U} \in \mathcal{S}^{ar} \times \mathcal{S}^{\bar{a}r} \times \mathcal{S}^{a\bar{r}}$ as its preimage. Due to $(s^{ar*}, s^{\bar{a}r*'}, s^{a\bar{r}*}) \in \mathcal{U}$, $\mathcal{U}$ is nonempty. Further, as $\mathcal{U}$ is nonempty and open, $\mathcal{U}$ has a Lebesgue measure of greater than zero.

As we assume that $p_{s^{ar}, s^{\bar{a}r}, s^{a\bar{r}}}$ is fully supported over the entire domain $\mathcal{S}^{ar} \times \mathcal{S}^{\bar{a}r} \times \mathcal{S}^{a\bar{r}}$, we can deduce that $\mathrm{P}_p[\mathcal{U}] > 0$. That is, $\psi > 0$ with a probability greater than zero, which contradicts the invariance condition, Therefore, we can show that $\hat{h}_r(s^r)$ does not depend on $s^{\bar{a}r}$.

Similarly, we can show that $\hat{h}_a(s^a)$ does not depend on $s^{a\bar{r}}$.

Finally, the smoothness and invertibility of $\hat{h}_r$ and $\hat{h}_a$ follow from the smoothness and invertibility of $h_r$ and $h_a$ over the entire domain.

Therefore, $h_r(h_a)$ is a smooth invertible mapping between $s^{ar}$ and $\hat{s}^{ar}$. That is, $s^{ar}$ is blockwise invertible.

**Step 4: prove the blockwise identifiability of $s_t^{\bar{a}r}$, $s_t^{a\bar{r}}$, and $s_t^{\bar{a}\bar{r}}$.**

We can use the same technique in Step 3 to show the identifiability of $s^{\bar{a}r}$ and $s^{a\bar{r}}$. Specifically, since $s_r$ and $s^{ar}$ are identifiable, we can show that $s^{\bar{a}r}$ is identifiable. Similarly, since $s^a$ and $s^{ar}$ are identifiable, we can show that $s^{a\bar{r}}$ is identifiable. Furthermore, since $s^{ar}$, $s^{\bar{a}r}$, and $s^{a\bar{r}}$ are identifiable, we can show that $s^{\bar{a}\bar{r}}$ is identifiable □

# B  Derivation of the Objective Function

We start by defining the components of the world mode as follows:

$$
\begin{cases}
\textit{Observation Model:} & p_\theta(o_t \mid \mathbf{s_t}) \\
\textit{Reward Model:} & p_\theta(r_t \mid s_t^r) \\
\textit{Transition Model:} & p_\gamma(\mathbf{s_t} \mid \mathbf{s_{t-1}}, a_{t-1}) \\
\textit{Representation Model:} & q_\phi(\mathbf{s_t} \mid o_t, \mathbf{s_{t-1}}, a_{t-1})
\end{cases}
\tag{25}
$$

The latent dynamics can be disentangled into four catogories:

$$
\begin{array}{ll}
\textit{Disentangled Transition Model:} & \textit{Disentangled Representation Model:} \\[4pt]
\begin{cases}
p_{\gamma_1}(s_t^{ar} \mid s_{t-1}^r, a_{t-1}) \\
p_{\gamma_2}(s_t^{\bar{a}r} \mid s_{t-1}^r) \\
p_{\gamma_3}(s_t^{a\bar{r}} \mid \mathbf{s_{t-1}}, a_{t-1}) \\
p_{\gamma_4}(s_t^{\bar{a}\bar{r}} \mid \mathbf{s_{t-1}})
\end{cases}
&
\begin{cases}
q_{\phi_1}(s_t^{ar} \mid o_t, s_{t-1}^r, a_{t-1}) \\
q_{\phi_2}(s_t^{\bar{a}r} \mid o_t, s_{t-1}^r) \\
q_{\phi_3}(s_t^{a\bar{r}} \mid o_t, \mathbf{s_{t-1}}, a_{t-1}) \\
q_{\phi_4}(s_t^{\bar{a}\bar{r}} \mid o_t, \mathbf{s_{t-1}})
\end{cases}
\end{array}
\tag{26}
$$

We follow the derivation framework in Dreamer [6] and define the information bottleneck objective for latent dynamics models [25]

$$
\max I\left(\mathbf{s_{1:T}}; (o_{1:T}, r_{1:T}) \mid a_{1:T}\right) - \beta \cdot I\left(\mathbf{s_{1:T}}, i_{1:T} \mid a_{1:T}\right),
\tag{27}
$$

where $\beta$ is scalar and $i_t$ are dataset indices that determine the observations $p(o_t \mid i_t) = \delta(o_t - \bar{o}_t)$ as in [26]. Maximizing the objective leads to model states that can predict the sequence of observations and rewards while limiting the amount of information extracted at each time step. We derive the lower bound of the first term in Equation 27:

$$
\begin{aligned}
& I\left(\mathbf{s_{1:T}}; (o_{1:T}, r_{1:T}) \mid a_{1:T}\right) \\
&= E_{q(o_{1:T}, r_{1:T}, \mathbf{s_{1:T}}, a_{1:T})}\left(\sum_t \ln p\left(o_{1:T}, r_{1:T} \mid \mathbf{s_{1:T}}, a_{1:T}\right) - \underbrace{\ln p\left(o_{1:T}, r_{1:T} \mid a_{1:T}\right)\}}_{\text{const}}\right) \\
&\overset{+}{=} E\left(\sum_t \ln p\left(o_{1:T}, r_{1:T} \mid \mathbf{s_{1:T}}, a_{1:T}\right)\right) \\
&\geq E\left(\sum_t \ln p\left(o_{1:T}, r_{1:T} \mid \mathbf{s_{1:T}}, a_{1:T}\right)\right) - \mathrm{KL}\left(p\left(o_{1:T}, r_{1:T} \mid \mathbf{s_{1:T}}, a_{1:T}\right) \| \prod_t p_\theta\left(o_t \mid s_t\right) p_\theta\left(r_t \mid s_t^r\right)\right) \\
&= E\left(\sum_t \ln p_\theta\left(o_t \mid \mathbf{s_t}\right) + \ln p_\theta\left(r_t \mid s_t^r\right)\right).
\end{aligned}
\tag{28}
$$

Thus, we obtain the objective function:

$$
\mathcal{J}_O^t = \ln p_\theta\left(o_t \mid \mathbf{s_t}\right) \quad \mathcal{J}_R^t = \ln p_\theta\left(r_t \mid s_t^r\right)
\tag{29}
$$

For the second term in Equation 27, we use the non-negativity of the KL divergence to obtain an upper bound,

$$
\begin{aligned}
& I\left(\mathbf{s_{1:T}}; i_{1:T} \mid a_{1:T}\right) \\
&= E_{q(o_{1:T}, r_{1:T}, \mathbf{s_{1:T}}, a_{1:T}, i_{1:T})}\left(\sum_t \ln q\left(\mathbf{s_t} \mid \mathbf{s_{t-1}}, a_{t-1}, i_t\right) - \ln p\left(\mathbf{s_t} \mid \mathbf{s_{t-1}}, a_{t-1}\right)\right) \\
&= E\left(\sum_t \ln q_\phi\left(\mathbf{s_t} \mid \mathbf{s_{t-1}}, a_{t-1}, o_t\right) - \ln p\left(\mathbf{s_t} \mid \mathbf{s_{t-1}}, a_{t-1}\right)\right) \\
&\leq E\left(\sum_t \ln q_\phi\left(\mathbf{s_t} \mid \mathbf{s_{t-1}}, a_{t-1}, o_t\right) - \ln p_\gamma\left(\mathbf{s_t} \mid \mathbf{s_{t-1}}, a_{t-1}\right)\right) \\
&= E\left(\sum_t \mathrm{KL}\left(q_\phi\left(\mathbf{s_t} \mid \mathbf{s_{t-1}}, a_{t-1}, o_t\right) \| p_\gamma\left(\mathbf{s_t} \mid \mathbf{s_{t-1}}, a_{t-1}\right)\right)\right).
\end{aligned}
\tag{30}
$$

According to equation 26, we have $p_\gamma = p_{\gamma_1} \cdot p_{\gamma_2} \cdot p_{\gamma_3} \cdot p_{\gamma_4}$ and $q_\phi = q_{\phi_1} \cdot q_{\phi_2} \cdot q_{\phi_3} \cdot q_{\phi_4}$.

$$
\begin{aligned}
\mathrm{KL}(q_\phi \| p_\gamma) &= \mathrm{KL}(q_{\phi_1} \cdot q_{\phi_2} \cdot q_{\phi_3} \cdot q_{\phi_4} \| p_{\gamma_1} \cdot p_{\gamma_2} \cdot p_{\gamma_3} \cdot p_{\gamma_4}) \\
&= E_{q_\phi} \left( \ln \frac{q_{\phi_1}}{p_{\gamma_1}} + \ln \frac{q_{\phi_2}}{p_{\gamma_2}} + \ln \frac{q_{\phi_3}}{p_{\gamma_3}} + \ln \frac{q_{\phi_4}}{p_{\gamma_4}} \right) \\
&= \mathrm{KL}(q_{\phi_1} \| p_{\gamma_1}) + \mathrm{KL}(q_{\phi_2} \| p_{\gamma_2}) + \mathrm{KL}(q_{\phi_3} \| p_{\gamma_3}) + \mathrm{KL}(q_{\phi_4} \| p_{\gamma_4})
\end{aligned}
\tag{31}
$$

We introduce additional hyperparameters to regulate the amount of information contained within each category of variables:

$$
\mathcal{J}_{\mathrm{D}}^t = -\beta_1 \cdot \mathrm{KL}\left(q_{\phi_1} \| p_{\gamma_1}\right) - \beta_2 \cdot \mathrm{KL}\left(q_{\phi_2} \| p_{\gamma_2}\right) - \beta_3 \cdot \mathrm{KL}\left(q_{\phi_3} \| p_{\gamma_3}\right) - \beta_4 \cdot \mathrm{KL}\left(q_{\phi_4} \| p_{\gamma_4}\right). \tag{32}
$$

Additionally, we introduce two supplementary objectives to explicitly capture the distinctive characteristics of the four distinct representation categories. Specifically, we characterize the reward-relevant representations by measuring the dependence between $s_t^r$ and $R_t$, given $a_{t-1:t}$ and $s_{t-1}^r$, that is $I(s_t^r, R_t \mid a_{t-1:t}, s_{t-1}^r)$ (see Figure 15(a). Note that if the action $a_t$ is not dependent on $s_t^r$, such as when it is randomly chosed. $a_t$ does not need to be conditioned. In this case, the mutual information turns into $I(s_t^r, R_t \mid a_{t-1}, s_{t-1}^r)$. To ensure that $s_t^r$ are minimally sufficient for policy training, we maximize $I(s_t^r, R_t \mid a_{t-1:t}, s_{t-1}^r)$ while minimizing $I(s_t^{\bar{r}}, R_t \mid a_{t-1:t}, s_{t-1}^r)$ to discourage the inclusion of redundant information in $s_t^{\bar{r}}$ concerning the rewards:

$$
I(s_t^r; R_t \mid a_{t-1:t}, s_{t-1}^r) - I(s_t^{\bar{r}}; R_t \mid a_{t-1:t}, s_{t-1}^r). \tag{33}
$$

The conditional mutual information can be expressed as the disparity between two mutual information values.

$$
\begin{aligned}
I(s_t^r; R_t \mid a_{t-1:t}, s_{t-1}^r) &= I(R_t; s_t^r, a_{t-1:t}, s_{t-1}^r) - I(R_t; a_{t-1:t}, s_{t-1}^r), \\
I(s_t^{\bar{r}}; R_t \mid a_{t-1:t}, s_{t-1}^r) &= I(R_t; s_t^{\bar{r}}, a_{t-1:t}, s_{t-1}^r) - I(R_t; a_{t-1:t}, s_{t-1}^r).
\end{aligned}
\tag{34}
$$

Combining the above two equations, we eliminated the identical terms, ultimately yielding the following formula

$$
I(R_t; s_t^r, a_{t-1:t}, s_{t-1}^r) - I(R_t; s_t^{\bar{r}}, a_{t-1:t}, s_{t-1}^r). \tag{35}
$$

We use the Donsker-Varadhan representation to express mutual information as a supremum over functions,

$$
\begin{aligned}
I(X; Y) &= D_{KL}(p(x,y) \| p(x)p(y)) \\
&= \sup_{T \in \mathcal{T}} \mathbb{E}_{p(x,y)}[T(x,y)] - \log \mathbb{E}_{p(x)p(y)}[e^{T(x,y)}].
\end{aligned}
\tag{36}
$$

We employ mutual information neural estimation [27] to approximate the mutual information value. We represent the function $T$ using a neural network that accepts variables $(x, y)$ as inputs and is parameterized by $\alpha$. The neural network is optimized through stochastic gradient ascent to find the supremum. Substituting $x$ and $y$ with variables defined in Equation 35, our objective is reformulated as follows:

$$
\mathcal{J}_{\mathrm{RS}}^t = \lambda_1 \cdot \{I_{\alpha_1}(R_t; s_t^r, a_{t-1:t}, \mathbf{sg}(s_{t-1}^r)) - I_{\alpha_2}(R_t; s_t^{\bar{r}}, a_{t-1:t}, \mathbf{sg}(s_{t-1}^r))\}. \tag{37}
$$

To incorporate the conditions from the original objective, we apply the stop_gradient operation to the variable $s_{t-1}^r$. Similarly, to ensure that the representations $s_t^a$ are directly controllable by actions, while $s_t^{\bar{a}}$ are not, we maximize the following objective:

$$
I(s_t^a; a_{t-1} \mid \mathbf{s_{t-1}}) - I(s_t^{\bar{a}}, a_{t-1} \mid \mathbf{s_{t-1}}), \tag{38}
$$

By splitting the conditional mutual information and eliminating identical terms, we obtain the following objective function:

$$
\mathcal{J}_{\mathrm{AS}}^t = \lambda_2 \cdot \{I_{\alpha_3}(a_{t-1}; s_t^a, \mathbf{sg}(\mathbf{s_{t-1}})) - I_{\alpha_4}(a_{t-1}; s_t^{\bar{a}}, \mathbf{sg}(\mathbf{s_{t-1}}))\}. \tag{39}
$$

where $\alpha_1, \alpha_2, \alpha_3, \alpha_4$ can be obtained by maximizing Equation 36. Intuitively, these two objective functions ensure that $s_t^r$ is predictive of the reward, while $s_t^{\bar{r}}$ is not; similarly, $s_t^a$ can be predicted by the action, whereas $s_t^{\bar{a}}$ cannot.

Combine the equation 29, equation 32, equation 37 and equation 39, the total objective function is:

$$
\begin{aligned}
\mathcal{J}_{\text{TOTAL}} &= \max_{\phi,\theta,\gamma,\alpha_1,\alpha_3} \min_{\alpha_2\alpha_4} \mathrm{E}_{q_\phi}\left( \sum_t \left( \mathcal{J}_O^t + \mathcal{J}_R^t + \mathcal{J}_D^t + \mathcal{J}_{\text{RS}}^t + \mathcal{J}_{\text{AS}}^t \right) \right) + \text{const} \\
&= \max_{\phi,\theta,\gamma,\alpha_1,\alpha_3} \min_{\alpha_2,\alpha_4} \mathbb{E}_{q_\phi}\{ \log p_\theta(o_t \mid \mathbf{s_t}) + \log p_\theta(r_t \mid s_t^r) \\
&\qquad - \sum_{i=1}^4 \beta_i \cdot \mathrm{KL}(q_{\phi_i} \| p_{\gamma_i}) + \lambda_1 \cdot (I_{\alpha_1} - I_{\alpha_2}) + \lambda_2 \cdot (I_{\alpha_3} - I_{\alpha_4})\} + \text{const} .
\end{aligned}
\tag{40}
$$

The expectation is computed over the dataset and the representation model. Throughout the model learning process, the objectives for estimating mutual information and learning the world model are alternately optimized.

## B.1  Discussions

In this subsection, we examine the mutual information constraints in equation 37 and equation 39 and their relationship with other objectives. Our findings reveal that while other objectives partially fulfill the desired functionality of the mutual information constraints, incorporating both mutual information objectives is essential for certain environments.

The objective functions can be summarized as follows::

$$
\begin{aligned}
\mathcal{J}_O^t &= \ln p_\theta\left(o_t \mid \mathbf{s_t}\right), \quad \mathcal{J}_R^t = \ln p_\theta\left(r_t \mid s_t^r\right), \quad \mathcal{J}_D^t = -\mathrm{KL}\left(q_\phi \| p_\gamma\right), \\
\mathcal{J}_{\text{RS}}^t &= \lambda_1 \cdot \{I_{\alpha_1}(R_t; s_t^r, a_{t-1:t}, \mathbf{sg}(s_{t-1}^r)) - I_{\alpha_2}(R_t; s_t^{\bar{r}}, a_{t-1:t}, \mathbf{sg}(s_{t-1}^r))\}, \\
\mathcal{J}_{\text{AS}}^t &= \lambda_2 \cdot \{I_{\alpha_3}(a_{t-1}; s_t^a, \mathbf{sg}(\mathbf{s_{t-1}})) - I_{\alpha_4}(a_{t-1}; s_t^{\bar{a}}, \mathbf{sg}(\mathbf{s_{t-1}}))\},
\end{aligned}
\tag{41}
$$

and the KL divergence term can be further decomposed into 4 components:

$$
\begin{aligned}
\mathcal{J}_{D_1}^t &= -\beta_1 \cdot \mathrm{KL}(q_{\phi_1}\left(s_t^{ar} \mid o_t, s_{t-1}^r, a_{t-1}\right) \| p_{\gamma_1}\left(s_t^{ar} \mid s_{t-1}^r, a_{t-1}\right)) \\
\mathcal{J}_{D_2}^t &= -\beta_2 \cdot \mathrm{KL}(q_{\phi_2}\left(s_t^{\bar{a}r} \mid o_t, s_{t-1}^r\right) \| p_{\gamma_2}\left(s_t^{\bar{a}r} \mid s_{t-1}^r\right)) \\
\mathcal{J}_{D_3}^t &= -\beta_3 \cdot \mathrm{KL}(q_{\phi_3}\left(s_t^{a\bar{r}} \mid o_t, \mathbf{s_{t-1}}, a_{t-1}\right) \| p_{\gamma_3}\left(s_t^{a\bar{r}} \mid \mathbf{s_{t-1}}, a_{t-1}\right)) \\
\mathcal{J}_{D_4}^t &= -\beta_4 \cdot \mathrm{KL}(q_{\phi_4}\left(s_t^{\bar{a}\bar{r}} \mid o_t, \mathbf{s_{t-1}}\right) \| p_{\gamma_4}\left(s_t^{\bar{a}\bar{r}} \mid \mathbf{s_{t-1}}\right)).
\end{aligned}
\tag{42}
$$

Specifically, maximizing $I_{\alpha_1}$ in $\mathcal{J}_{\text{RS}}^t$ enhances the predictability of $R_t$ based on the current state $s_{t-1}^r$ conditioning on $(s_{t-1}^r, a_{t-1:t})$. However, notice that this objective can be partially accomplished by optimizing $\mathcal{J}_R^t$. When learning the world model, both the transition function and the reward function are trained: the reward function predicts the current reward $r_t$ using $s_t^r$, while the transition model predicts the next state. These combined predictions contribute to the overall prediction of $R_t$.

Minimizing $I_{\alpha_2}$ in $\mathcal{J}_{\text{RS}}^t$ eliminates extraneous reward-related information present in $s_t^{\bar{r}}$. According to our formulation, $s_t^{\bar{r}}$ can still be predictive of $R_t$ as long as it does not introduce additional predictability beyond what is already captured by $(s_{t-1}^r, a_{t-1:t})$. This is because we only assume that $s_t^{\bar{r}}$ is **conditionally** independent from $R_t$ when conditioning on $(s_{t-1}^r, a_{t-1:t})$. If we don't condition on $(s_{t-1}^r, a_{t-1:t})$, it introduces $s_{t-1}^{\bar{r}}$ as the confounding factor between $s_t^{\bar{r}}$ and $R_t$, establishing association between $s_t^{\bar{r}}$ and $R_t$ (refer to Figure 15(a)). Note that the KL divergence constraints govern the information amount within each state category. By amplifying the weight of the KL constraints on $s_t^{\bar{r}}$, the value of $I_{\alpha_2}$ can indirectly be diminished.

By maximizing $I_{\alpha_3}$ and minimizing $I_{\alpha_4}$ in $\mathcal{J}_{\text{AS}}^t$ , we ensure that $s_t^a$ can be predicted based on $(a_{t-1}, \mathbf{s_{t-1}})$ while $s_t^{\bar{a}}$ cannot. The KL constraints on $s_t^{ar}$ and $s_t^{a\bar{r}}$ incorporate the action $a_{t-1}$ into the prior and posterior, implicitly requiring that $s_t^a$ should be predictable given $a_{t-1}$. Conversely, the KL constraints on $s_t^{\bar{a}r}$ and $s_t^{\bar{a}\bar{r}}$ do not include the action $a_{t-1}$ in the prior and posterior, implicitly requiring that $s_t^a$ should not be predictable based on $a_{t-1}$. However, relying solely on indirect constraints can sometimes be ineffective, as it may lead to entangled representations that negatively impact policy performance (see Figure 13).

**Ablation of the mutual information constraints.**  The inclusion of both $\mathcal{J}_{\text{RS}}^t$ and $\mathcal{J}_{\text{AS}}^t$ is essential in certain environments to promote disentanglement and enhance policy performance, despite sharing

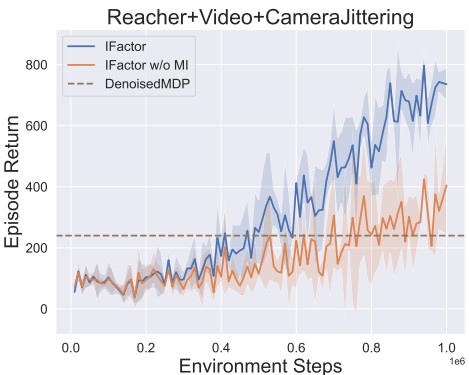

Figure 7: Ablation of the mutual information constraints in the Reacher environment with video background and jittering camera. The dashed brown line illustrates the policy performance of Denoised MDP after 1 million environment steps.

some common objectives. We have observed improved training stability in the variant of Robodesk environment (see Figure 12) and significant performance gains in the Reacher environment with video background and camera jittering (see Figure 7). When two mutual information objectives are removed, we notice that entangled representations emerged in these environments, as depicted in Figure 13. We assign values of 0.1 to $\lambda_1$ and $\lambda_2$ in the environment of modified Cartpole, variant of Robodesk and Reacher with video background and jittering camera. Empirically, a value of 0.1 has been found to be preferable for both $\lambda_1$ and $\lambda_2$. Using a higher value for regularization might negatively impact the learning of representation and transition model. In other DMC environments, the ELBO loss alone has proven effective due to the inherent structure of our disentangled latent dynamics. The choice of hyperparameters $(\beta_1, \beta_2, \beta_3, \beta_4)$ depends on the specific goals of representation learning and the extent of noise interference in the task. If the objective is to accurately recover the true latent variables for understanding the environment, it is often effective to assign equal weights to the four KL divergence terms (for experiments on synthetic data and modified cartpole). When the aim is to enhance policy training stability by mitigating noise, it is recommended to set the values of $\beta_1$ and $\beta_2$ higher than $\beta_3$ and $\beta_4$ (for experiments on variants of Robodesk and DMC). Moreover, in environments with higher levels of noise, it is advisable to increase the discrepancy in values between the hyperparameters.

## C   Algorithm

---

**Algorithm 1:** IFactor

---

**Input:**   Representation model: $q_\phi(s_t \mid s_{t-1}, a_{t-1}, o_t) = q_{\phi_1} \cdot q_{\phi_2} \cdot q_{\phi_3} \cdot q_{\phi_4}$;
   Transition model: $p_\gamma(s_t \mid s_{t-1}, a_{t-1}) = p_{\gamma_1} \cdot p_{\gamma_2} \cdot p_{\gamma_3} \cdot p_{\gamma_4}$;
   Observation Model: $p_\theta(o_t \mid s_t)$; Reward model: $p_\theta(r_t \mid s_t^r)$;
   Policy Model: $\pi_\psi(a_t \mid s_t^r)$;  Value model: $v_\psi(s_t^r)$;
   Mutual information estimator: $I_{\alpha_1}, I_{\alpha_2}, I_{\alpha_3}, I_{\alpha_4}$;
   Policy optimization algorithm PI-OPT, which is in default the same as that used in Dreamer.

**Output:**  Transition Model $p_\gamma$. Representation Model $q_\phi$. Policy Model $\pi_\psi$.

  1: **while** training **do**
  2:    // Exploration
  3:    Collect new trajectories $D' = \{(o_t, a_t, r_t)\}_t$ with $\pi$ acting on $q_\phi$ encoded outputs
  4:    Add experience to the replay buffer $D = D \cup D'$
  5:    // Model learning
  6:    Sample a batch of data sequences $\{(o_t, a_t, r_t)\}_t$ from the reply buffer $D$
  7:    Obtain $s_t$ by the representation model and estimate mutual information terms $I_{\alpha_1}, I_{\alpha_2}, I_{\alpha_3}, I_{\alpha_4}$
  8:    **for** $i = 1$ to $n$ **do**
  9:       Train mutual information neural estimators by maximizing $I_{\alpha_1}, I_{\alpha_2}, I_{\alpha_3}, I_{\alpha_4}$
 10:    **end for**
 11:    Freeze the parameters in $I_{\alpha_1}, I_{\alpha_2}, I_{\alpha_3}, I_{\alpha_4}$ and Train $q_\phi$, $p_\gamma$ and $p_\theta$ with Equation 11
 12:    // Policy learning by dreaming
 13:    Imagine trajectories of $s_t^r$ using the learned world model.
 14:    Train $\pi_\psi$ and $v_\psi$ by running PI-OPT
 15: **end while**

---

# D  Environment Descriptions

## D.1  Synthetic Data

For the sake of simplicity, we consider one lag for the latent processes in Section 4. Our identifiability proof can actually be applied for arbitrary lags directly because the identifiability does not rely on the number of previous states. We extend the latent dynamics of the synthetic environment to incorporate a general time-delayed causal effect with $\tau \geq 1$ in the synthetic environment. When $\tau = 1$, it reduces to a common MDP. The ground-truth generative model of the environment is as follows::

$$
\begin{cases}
\textit{Observation Model:} & p_\theta(o_t \mid \mathbf{s_t}) \\
\textit{Reward Model:} & p_\theta(r_t \mid s_t^r) \\
\textit{Transition Model:} & p_\gamma(\mathbf{s_t} \mid \mathbf{s_{t-\tau:t-1}}, a_{t-\tau:t-1})
\end{cases}
\qquad
\textit{Transition}:
\begin{cases}
p_{\gamma_1}\left(s_t^{ar} \mid s_{t-\tau:t-1}^r, a_{t-\tau:t-1}\right) \\
p_{\gamma_2}\left(s_t^{\bar{a}r} \mid s_{t-\tau:t-1}^r\right) \\
p_{\gamma_3}\left(s_t^{a\bar{r}} \mid \mathbf{s_{t-\tau:t-1}}, a_{t-\tau:t-1}\right) \\
p_{\gamma_4}\left(s_t^{\bar{a}\bar{r}} \mid \mathbf{s_{t-\tau:t-1}}\right)
\end{cases}
\tag{43}
$$

**Data Generation**  We generate synthetic datasets with $100,000$ data points according to the generating process in Equation 43, which satisfies the identifiability conditions stated in Theorem 1. The latent variables $\mathbf{s_t}$ have 8 dimensions, where $s_t^{ar} = s_t^{\bar{a}r} = s_t^{a\bar{r}} = s_t^{\bar{a}\bar{r}} = 2$. At each timestep, a one-hot action of dimension 5, denoted as $a_t$, is taken. The lag number of the process is set to $\tau = 2$. The observation model $p_\theta(o_t, \mid \mathbf{s_t})$ is implemented using a random three-layer MLPs with LeakyReLU units. The reward model $p_\theta(r_t, \mid s_t^r)$ is represented by a random one-layer MLP. It's worth noting that the reward model is not invertible due to the scalar nature of $r_t$. Four distinct transition functions, namely $p_{\gamma_1}$, $p_{\gamma_2}$, $p_{\gamma_3}$, and $p_{\gamma_4}$, are employed and modeled using random one-layer MLP with LeakyReLU units. The process noise is sampled from an i.i.d. Gaussian distribution with a standard deviation of $\sigma = 0.1$. To simulate nonstationary noise for various latent variables in RL, the process noise terms are coupled with the historical information by multiplying them with the average value of all the time-lagged latent variables, as suggested in [19].

## D.2  Modified Cartpole

We have modified the original Cartpole environment by introducing two distractors. The first distractor is an uncontrollable Cartpole located in the upper portion of the image, which does not affect the rewards. The second distractor is a controllable green light positioned below the reward-relevant Cartpole in the lower part of the image, but it is not associated with any rewards. The task-irrelevant cartpole undergoes random actions at each time step and stops moving when its angle exceeds 45 degrees or goes beyond the screen boundaries. The action space consists of three independent degrees of freedom: direction (left or right), force magnitude (10N or 20N), and green light intensity (lighter or darker). This results in an 8-dimensional one-hot vector. The objective of this variant is to maintain balance for the reward-relevant cartpole by applying suitable forces.

## D.3  Variant of Robodesk

The RoboDesk environment with noise distractors [9] is a control task designed to simulate realistic sources of noise, such as flickering lights and shaky cameras. Within the environment, there is a large TV that displays natural RGB videos. On the desk, there is a green button that controls both the hue of the TV and a light. The agent's objective is to manipulate this button in order to change the TV's hue to green. The agent's reward is determined based on the greenness of the TV image. In this environment, all four types of information are present (see Table 1).

| Modified Cartpole | Robodesk | Reacher Easy | Cheetah Run | Walker Walk |

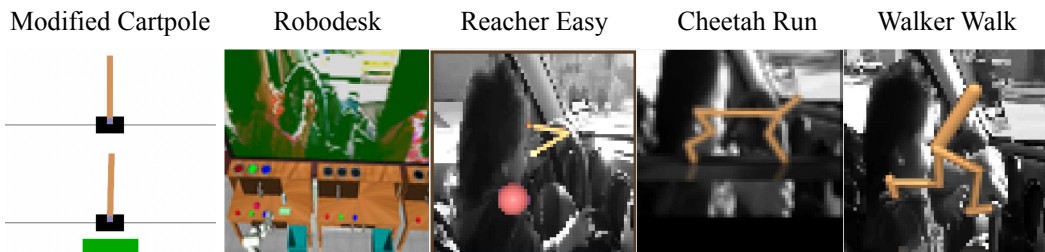

Figure 8: Visualization of the environments used in our experiments.

| | Ctrl + Rew | $\overline{\text{Ctrl}}$ + Rew | Ctrl + $\overline{\text{Rew}}$ | $\overline{\text{Ctrl}}$ + $\overline{\text{Rew}}$ |
|---|---|---|---|---|
| Modified Cartpole | Agent | Agent | Green Light | Distractor cartpole |
| Robodesk | Agent, Button, Light on desk | TV content, Button sensor noise | Blocks on desk, Handle on desk, Other movable objects **(Green hue of TV)** | Jittering and flickering environment lighting, Jittering camera |
| DMC **Noiseless** | Agent | **(Agent)** | — | — |
| **Video Background** | Agent | **(Agent)** | — | Background |
| **Video Background + Noisy Sensor** | Agent | **(Agent)** Background | — | — |
| **Video Background + Camera Jittering** | Agent | **(Agent)** | — | Background Jittering camera |

Table 1: Categorization of various types of information in the environments we evaluated. We use the red color to emphasize the categorization difference between IFactor and Denoised MDP. Unlike Denoised MDP that assumes independent latent processes, IFactor allows for causally-related processes. Therefore, in this paper, the term "controllable" refers specifically to one-step controllability, while "reward-relevant" is characterized by the conditional dependence between $s_t^*$ and the cumulative reward variable $R_t$ when conditioning on $(s_{t-1}, a_{t-1:t})$. Following this categorization, certain agent information can be classified as (one-step) uncontrollable (including indirectly controllable and uncontrollable factors) but reward-relevant factors, such as some position information determined by the position and velocity in the previous time-step rather than the action. On the other hand, the green hue of TV in Robodesk is classified as controllable but reward-irrelevant factors, as they are independent of the reward given the state of the robot arm and green button, aligning with the definition of $s_t^{a\bar{r}}$.

## D.4 Variants of DeepMind Control Suite

Four variants [9] are introduced for each DMC task:

- **Noiseless**: Original environment without distractors.

- **Video Background**: Replacing noiseless background with natural videos [37] ($\overline{\text{Ctrl}} + \overline{\text{Rew}}$).

- **Video Background + Sensor Noise**: Imperfect sensors sensitive to intensity of a background patch ($\overline{\text{Ctrl}} + \text{Rew}$).

- **Video Background + Camera Jittering**: Shifting the observation by a smooth random walk ($\overline{\text{Ctrl}} + \overline{\text{Rew}}$).

The video background in the environment incorporates grayscale videos from Kinetics-400, where pixels with high blue channel values are replaced. Camera jittering is introduced through a smooth random walk shift using Gaussian-perturbing acceleration, velocity decay, and pulling force. Sensor noise is added by perturbing a specific sensor based on the intensity of a patch in the background video. The perturbation involves adding the average patch value minus 0.5. Different sensors are perturbed for different environments. These sensor values undergo non-linear transformations, primarily piece-wise linear, to compute rewards. While the additive reward noise model may not capture sensor behavior perfectly, it is generally sufficient as long as the values remain within moderate ranges and stay within one linear region. (Note: the variants of Robodesk and DMC are not the contributions of this paper. We kindly refer readers to the paper of Denoised MDP [9] for a more detailed introduction.)

# E    Experimental Details

**Computing Hardware**    We used a machine with the following CPU specifications: Intel(R) Xeon(R) Silver 4110 CPU @ 2.10GHz; 32 CPUs, eight physical cores per CPU, a total of 256 logical CPU units. The machine has two GeForce RTX 2080 Ti GPUs with 11GB GPU memory.

**Reproducibility**    We've included the code for the framework and all experiments in the supplement. We plan to release our code under the MIT License after the paper review period.

## E.1    Synthetic Dataset

**Hyperparameter Selection and Network Structure**    We adopt a similar experimental setup to TDRL [19], while extending it by decomposing the dynamics into four causally related latent processes proposed in this paper (refer to Equation 43). For all experiments, we assign $\beta_1 = \beta_2 = \beta_3 = \beta_4 = 0.003$ as the weights for the KL divergence terms. In this particular experiment, we set $\lambda_1$ and $\lambda_2$ to 0 because the utilization of the ELBO loss alone has effectively maximized $J_{RS}^t$ and $J_{AS}^t$, as illustrated in Figure 10. Here, $J_{RS}^t$ represents $I_{\alpha_1} - I_{\alpha_2}$, and $J_{AS}^t$ represents $I_{\alpha_3} - I_{\alpha_4}$. The network structure employed in this experiment is presented in Table 2.

**Training Details**    The models are implemented in PyTorch 1.13.1. The VAE network is trained using AdamW optimizer for 100 epochs. A learning rate of 0.001 and a mini-batch size of 64 are used. We have used three random seeds in each experiment and reported the mean performance with standard deviation averaged across random seeds.

### E.1.1    The identifiability socres for baselines

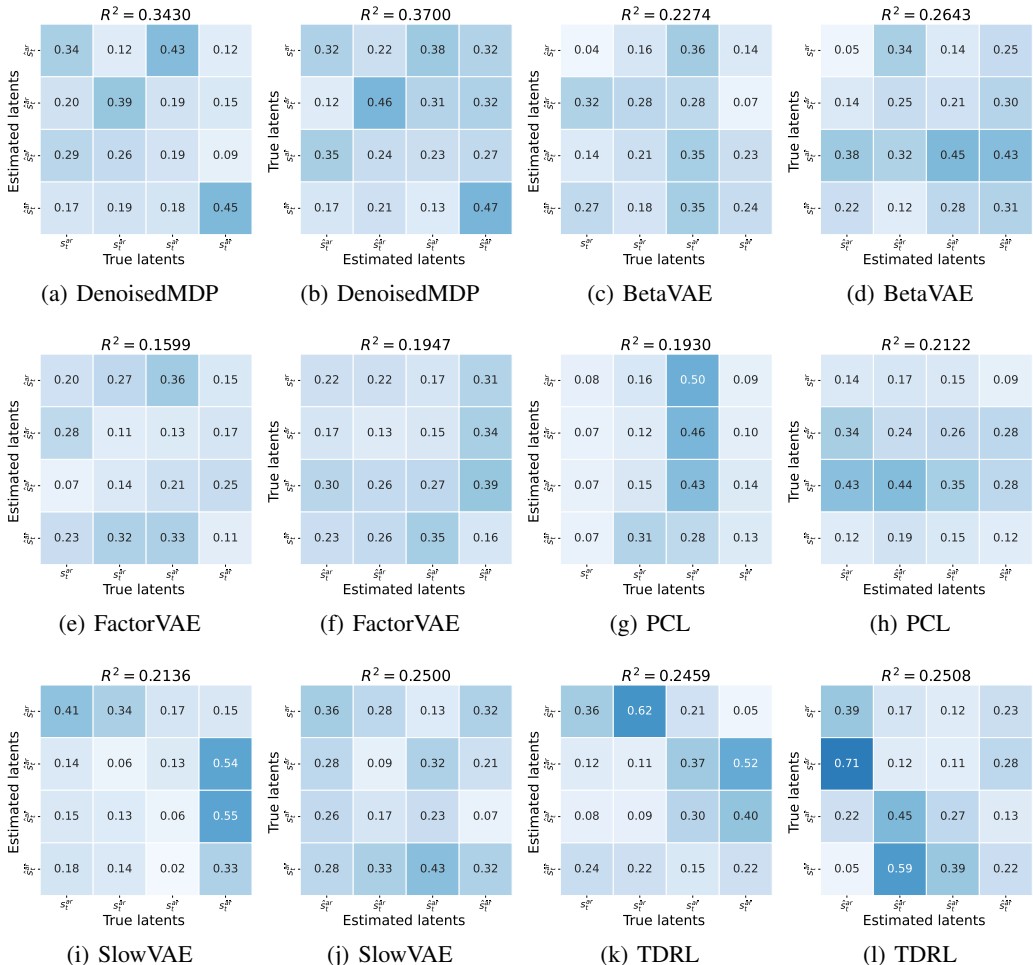

Figure 9: The identifiability socres for baselines in the experiments on synthetic data set. It can be observed that all baselines do not enjoy the property of block-wise identifiability.

Table 2: Architecture details. BS: batch size, T: length of time series, o_dim: observation dimension, s_dim: latent dimension, $s_t^{ar}$_dim: latent dimension for $s_t^{ar}$, $s_t^{\bar{a}r}$_dim: latent dimension for $s_t^{\bar{a}r}$, $s_t^{a\bar{r}}$_dim: latent dimension for $s_t^{a\bar{r}}$, $s_t^{\bar{a}\bar{r}}$_dim: latent dimension for $s_t^{\bar{a}\bar{r}}$ ( s_dim = $s_t^{ar}$_dim + $s_t^{\bar{a}r}$_dim + $s_t^{a\bar{r}}$_dim + $s_t^{\bar{a}\bar{r}}$_dim ), LeakyReLU: Leaky Rectified Linear Unit.

| Configuration | Description | Output |
|---|---|---|
| **1. MLP-Obs-Encoder** | Observation Encoder for Synthetic Data | |
| Input: $o_{1:T}$ | Observed time series | BS × T × o_dim |
| Dense | 128 neurons, LeakyReLU | BS × T × 128 |
| Dense | 128 neurons, LeakyReLU | BS × T × 128 |
| Dense | 128 neurons, LeakyReLU | BS × T × 128 |
| Dense | Temporal embeddings | BS × T × s_dim |
| **2. MLP-Obs-Decoder** | Observation Decoder for Synthetic Data | |
| Input: $\hat{s}_{1:T}$ | Sampled latent variables | BS × T × s_dim |
| Dense | 128 neurons, LeakyReLU | BS × T × 128 |
| Dense | 128 neurons, LeakyReLU | BS × T × 128 |
| Dense | o_dim neurons, reconstructed $\hat{\mathbf{o}}_{1:T}$ | BS × T × o_dim |
| **3. MLP-Reward-Decoder** | Reward Decoder for Synthetic Data | |
| Input: $\hat{s}_{1:T}$ | Sampled latent variables | BS × T × s_dim |
| Dense | 1 neurons, LeakyReLU | BS × T × 1 |
| **4. Disentangled Prior for $s_t^{ar}$** | Nonlinear Transition Prior Network | |
| Input | Sampled latents and actions $s_{1:T}^r, a_{1:T}$ | BS × T ×( $s_t^r$_dim + a_dim) |
| Dense | $s_t^{ar}$_dim neurons, prior output | BS × T × $s_t^{ar}$_dim |
| **5. Disentangled Prior for $s_t^{\bar{a}r}$** | Nonlinear Transition Prior Network | |
| Input | Sampled latent variable sequence $s_{1:T}^r$ | BS × T × $s_t^r$_dim |
| Dense | $s_t^{\bar{a}r}$_dim neurons, prior output | BS × T × $s_t^{\bar{a}r}$_dim |
| **6. Disentangled Prior for $s_t^{a\bar{r}}$** | Nonlinear Transition Prior Network | |
| Input | Sampled latents and actions $\mathbf{s_{1:T}}, a_{1:T}$ | BS × T × (s_dim + a_dim) |
| Dense | $s_t^{a\bar{r}}$_dim neurons, prior output | BS × T × $s_t^{a\bar{r}}$_dim |
| **7. Disentangled Prior for $s_t^{\bar{a}\bar{r}}$** | Nonlinear Transition Prior Network | |
| Input | Sampled latent variable sequence $\mathbf{s_{1:T}}$ | BS × T × s_dim |
| Dense | $s_t^{\bar{a}\bar{r}}$_dim neurons, prior output | BS × T × $s_t^{\bar{a}\bar{r}}$_dim |

### E.1.2 Extra Results.

During the training process, we record the estimation value of four mutual information (MI) terms. The corresponding results are presented in Figure 10. Despite not being explicitly incorporated into the objective function, the terms $I_{\alpha_1} - I_{\alpha_2}$ and $I_{\alpha_3} - I_{\alpha_4}$ exhibit significant maximization. Furthermore, the estimation values of $I_{\alpha_2}$ and $I_{\alpha_4}$ are found to be close to 0. These findings indicate that the state variable $s_t^{\bar{r}}$ contains little information about the reward, and the predictability of $s_t^{\bar{a}}$ by the action is also low.

### E.2 Modified Cartpole

In the modified Cartpole environment, we configure the values as follows: $\beta_1 = \beta_2 = \beta_3 = \beta_4 = 0.1$ and $\lambda_1 = \lambda_2 = 0.1$. Recurrent State Space Model (RSSM) uses a deterministic part and a stochastic part to represent latent variables. The deterministic state size for four dynamics are set to be (15, 15, 15, 15), and the stochastic state size are set to be (2, 2, 1, 4). The architecture of the encoder and decoder for observation is shown in Table 3 and Table 4 (64 × 64 resolution). Reward model uses 3-layer MLPs with hidden size to be 100 and four mutual information neural estimators are 4-layer MLPs with hidden size to be 128. Unlike the synthetic dataset, where there are clear categories

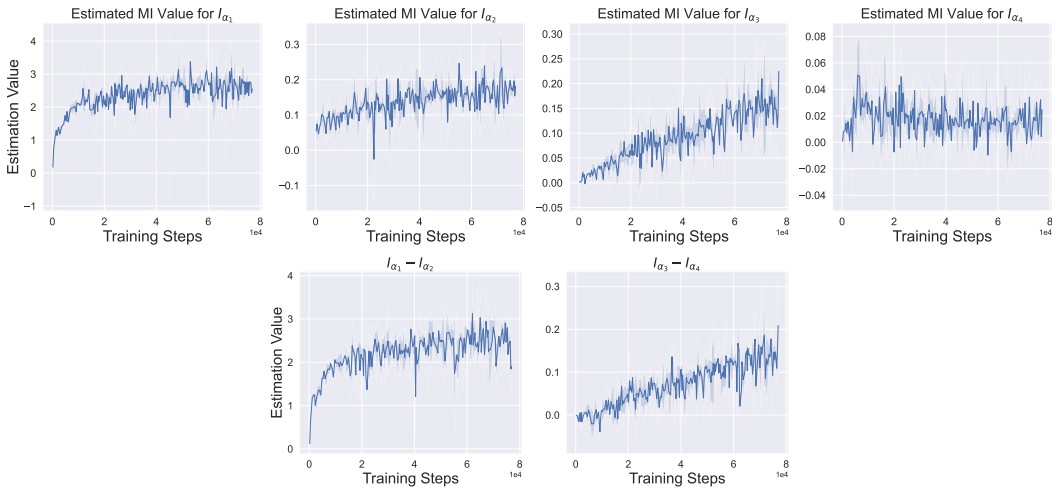

Figure 10: Estimation of the value of four mutual information terms and their differences in experiments on synthetic data.

.

for latent state variables, real-world situations pose a challenge due to the potential ambiguity in categorizing true variables. In our modified Cartpole environment, we defined the ground-truth $s_t^{ar}$ as the cart's position, $s_t^{\bar{a}r}$ as the pole's angle, $s_t^{a\bar{r}}$ as the light's greenness, and $s_t^{\bar{a}\bar{r}}$ as the distractor Cartpole's state (including cart position and pole angle). The disentanglement scores for individual $s_t^{ar}$ and $s_t^{\bar{a}r}$, as well as the combined $s_t^r$, are shown in Figure 11. We can obviously see that the true latent variables can be clearly identified.

| Operator | Input Shape | Kernel Size | Stride | Padding |
|---|---|---|---|---|
| Input | $[3, 96, 96]$ | — | — | — |
| Conv. + ReLU | $[32, 47, 47]$ | 4 | 2 | 0 |
| Conv. + ReLU | $[64, 22, 22]$ | 4 | 2 | 0 |
| Conv. + ReLU | $[128, 10, 10]$ | 4 | 2 | 0 |
| Conv. + ReLU | $[256, 4, 4]$ | 4 | 2 | 0 |
| Conv. + ReLU * | $[256, 2, 2]$ | 3 | 1 | 0 |
| Reshape + FC | $[1024]$ | — | — | — |

Table 3: The encoder architecture designed for observation resolution of $(96 \times 96)$. Its output is then fed into other networks for posterior inference. The default activation function used in the network is RELU. For observations with a resolution of $(64 \times 64)$, the last convolutional layer(*) is removed.

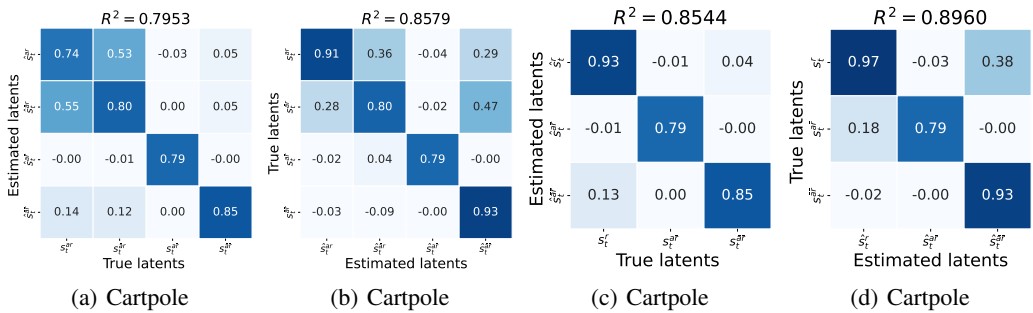

Figure 11: The identifiability socres for IFactor in the experiments on cartpole.

| Operator | Input Shape | Kernel Size | Stride | Padding |
|---|---|---|---|---|
| Input | [input_size] | — | — | — |
| FC + ReLU + Reshape | [1024, 1, 1] | — | — | — |
| Conv. Transpose + ReLU * | [128, 3, 3] | 3 | 1 | 0 |
| Conv. Transpose + ReLU | [128, 9, 9] | 5 | 2 | 0 |
| Conv. Transpose + ReLU | [64, 21, 21] | 5 | 2 | 0 |
| Conv. Transpose + ReLU | [32, 46, 46] | 6 | 2 | 0 |
| Conv. Transpose + ReLU | [3, 96, 96] | 6 | 2 | 0 |

Table 4: The decoder architecture designed for $(96 \times 96)$-resolution observation. For $(64 \times 64)$-resolution observation, the first transpose convolutional layer(*) is removed.

### E.3 Variant of Robodesk

In the variant of Robodesk, we conduct experiments with the following hyperparameter settings: $\beta_1 = \beta_2 = 2$, $\beta_3 = \beta_4 = 0.25$, and $\lambda_1 = \lambda_2 = 0.1$. For the four dynamics, we set the deterministic state sizes to (120, 40, 40, 40), and the stochastic state sizes to (30, 10, 10, 10). Denoised MDP utilizes two latent processes with deterministic state sizes [120, 120] and stochastic state sizes [20, 10]. For the mutual information neural estimators, we employ 4-layer MLPs with a hidden size of 128. To ensure a fair comparison, we align the remaining hyperparameters and network structure with those in the Denoised MDP. We reproduce the results of the Denoised MDP using their released code, maintaining consistency with their paper by employing the default hyperparameters. In order to evaluate the impact of the Mutual Information (MI) constraints, we conduct an ablation study. The results are shown is Figure 12. The constraints $\mathcal{J}_{RS}^t$ and $\mathcal{J}_{AS}^t$ are observed to stabilize the training process of IFactor. The results of IFactor are areaveraged over 5 runs, while the results of Denoised MDP and IFactor without MI are averaged over three runs.

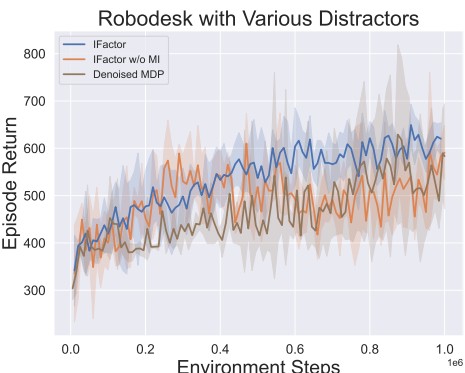

Figure 12: Comparison between IFactor and Denoised MDP in the variant of Robodesk environment.

**Policy learning based on the learned representations by IFactor**  We retrain policies using the Soft Actor-Critic algorithm [40] with various combinations of the four learned latent categories as input. We wrap the original environment with visual output using our representation model to obtain compact features. In this process, both deterministic states and stochastic states are utilized to form the feature. For instance, when referring to $s_t^r$, we use both the deterministic states and stochastic states of $s_t^r$. The implementation of SAC algorithm is based on Stableb-Baselines3[59], with a learning rate of 0.0002. Both the policy network and Q network consist of 4-layer MLPs with a hidden size of 256. We use the default hyperparameter settings in Stable-Baselines3 for other parameters.

## E.4 Variants of Deep Mind Control Suite

In the noiseless DMC environments, we set $\beta_1 = \beta_2 = \beta_3 = \beta_4 = 1$. For the DMC environments with video background, we set $\beta_1 = \beta_2 = 1$ and $\beta_3 = \beta_4 = 0.25$. In the DMC environments with video background and noisy sensor, we set $\beta_1 = \beta_2 = 2$ and $\beta_3 = \beta_4 = 0.25$. Lastly, for the DMC environments with video background and jittering camera, we set $\beta_1 = \beta_2 = 1$ and $\beta_3 = \beta_4 = 0.25$. Regarding the Reacher environment with video background and jittering camera, we set $\lambda_1 = \lambda_2 = 0.1$ for our experiments. For the other environments, we set $\lambda_1 = \lambda_2 = 0$. The deterministic state sizes for the four dynamics are set to (120, 120, 60, 60), while the stochastic state sizes are set to (20, 20, 10, 10). The four mutual information neural estimators utilize a 4-layer MLPs with a hidden size of 128. We align the other hyperparameters and network structure with those used in the Denoised MDP for a fair comparison.

| | Environment Steps | Action Repeat | Train Every | Collection Intervals | Batch Size | Sequence Length | Horizon |
|---|---|---|---|---|---|---|---|
| Modified Cartpole | 200,000 | 1 | 5 | 5 | 20 | 30 | 8 |
| Robodesk | 1,000,000 | 2 | 1000 | 100 | 50 | 50 | 15 |
| DMC | 1,000,000 | 2 | 1000 | 100 | 25 | 50 | 12 |

Table 5: Some hyperparameters of our method in the environment of Modified Cartpole, Robodesk and DMC. *Environment Steps* represents the number of interactions between the agent and the environment. *Action Repeat* determines how many times an agent repeats an action in a step. *Train Every* specifies the environment step between adjacent training iterations. *Collection Intervals* defines the number of times the model is trained in each training iteration (including world models, policy networks and value networks). *Batch Size* refers to the number of trajectories in each mini-batch. *Sequence Length* denotes the length of the chuck used in training the world models. *Horizon* determines the length of dreaming when training the policy using the world model. Hyperparameters are aligned with those used in the Denoised MDP for fair comparison.

### E.4.1 Policy optimization results on variants of DMC

We present detailed evaluation results in Table 6, showcasing both the mean values and the corresponding standard deviations for the final policy performance across each task. Results are averaged across three seeds. Denoised MDP performs well across all four variants with distinct noise types.

### E.4.2 Mutual information

Figure 7 demonstrates the notable improvement in policy performance in the Reacher environment with video background and jittering camera due to the inclusion of the constraints $J_{RS}^t$ and $J_{AS}^t$. To further investigate how they affects the model learning, we record the estimation values of four Mutual Information terms throughout the training process, as depicted in Figure 13. The results indicate that both $I_{\alpha_1} - I_{\alpha_2}$ and $I_{\alpha_3} - I_{\alpha_4}$ are maximized for both IFactor and IFactor without MI. However, IFactor exhibits a significantly higher rate of maximizing $I_{\alpha_3} - I_{\alpha_4}$ compared to IFactor without MI. This increased maximization leads to greater predictability of $s_t^a$ by the action, ultimately contributing to the observed performance gain.

### E.5 Visualization for DMC

In this experiment, we investigate five types of representations, which can be derived from the combination of four original disentangled representation categories. Specifically, $s_t^a$ is the controllable and reward relevant representation. $s_t^r = (s_t^{ar}, s_t^{\bar{a}r})$ is the reward-relevant representation. $s_t^{a\bar{r}}$ is the controllable but reward-irrelevant representation. $s_t^{\bar{a}\bar{r}}$ is the uncontrollable and reward-irrelevant representation (noise). $s_t^{\bar{r}} = (s_t^{a\bar{r}}, s_t^{\bar{a}\bar{r}})$ is the reward-irrelevant representation. Only representations of $s_t^r$ are used for policy optimization. We retrain 5 extra observation decoders to reconstruct the original image, which can precisely characterize what kind of information each type of representation contains, surpassing the limitations of the original decoder that is used in latent traversal. The visualization results are shown in Figure 14. It can be observed that $s_t^{ar}$ captures the movement of the agent partially but not well enough; $s_t^r$ captures the movement of the agent precisely but $s_t^{\bar{r}}$ fails (Reacher and Cheetah) or captures extra information of the background (Walker). This finding

| Cheetah | IFactor (Ours) | Dreamer Pro | Denoised MDP | Dreamer | TIA | DBC | CURL | PI-SAC |
|---|---|---|---|---|---|---|---|---|
| Noiseless | **874±39** | 803±75 | 771±60 | 714±348 | 766±29 | 182±50 | 171±34 | 369±28 |
| Video Background | **573±193** | 366±96 | 431±111 | 171±45 | 388±297 | 248±71 | 70±48 | 112±31 |
| Video Background + Noisy Sensor | **456±88** | 134±41 | 400±190 | 166±42 | 227±13 | 141±13 | 199±7 | 151±14 |
| Video Background + Camera Jittering | **418±22** | 150±104 | 294±100 | 160±32 | 202±93 | 141±35 | 169±22 | 156±20 |

| Walker | IFactor (Ours) | Dreamer Pro | Denoised MDP | Dreamer | TIA | DBC | CURL | PI-SAC |
|---|---|---|---|---|---|---|---|---|
| Noiseless | 966±5 | **941±16** | 947±13 | 955±6 | 955±5 | 614±111 | 417±296 | 203±92 |
| Video Background | **917±52** | 909±48 | 790±113 | 247±135 | 685±337 | 199±67 | 608±100 | 200±18 |
| Video Background + Noisy Sensor | **701±174** | 242±65 | 661±120 | 279±145 | 425±281 | 95±54 | 338±92 | 222±21 |
| Video Background + Camera Jittering | **524±194** | 368±301 | 291±104 | 106±22 | 230±332 | 62±17 | 448±70 | 116±6 |

| Reacher | IFactor (Ours) | Dreamer Pro | Denoised MDP | Dreamer | TIA | DBC | CURL | PI-SAC |
|---|---|---|---|---|---|---|---|---|
| Noiseless | **924±37** | 924±61 | 686±216 | 876±57 | 587±256 | 95±58 | 663±221 | 166±235 |
| Video Background | **963±10** | 555±92 | 544±121 | 253±127 | 123±21 | 102±58 | 751±189 | 76±35 |
| Video Background + Noisy Sensor | **839±51** | 675±137 | 561±182 | 202±82 | 264±280 | 97±39 | 607±260 | 85±5 |
| Video Background + Camera Jittering | **736±53** | 675±82 | 213±106 | 109±19 | 89±26 | 87±51 | 632±96 | 84±13 |

Table 6: Policy optimization evaluation on variants of DMC with various distractors. IFactor consistently performs well across all four variants with distinct noise types. Bold numbers show the best model-learning result for specific policy learning choices. Results are averaged over 3 runs.

suggests that $s_t^r$ contains sufficient information within the original noisy observation for effective control, while effectively excluding other sources of noise.

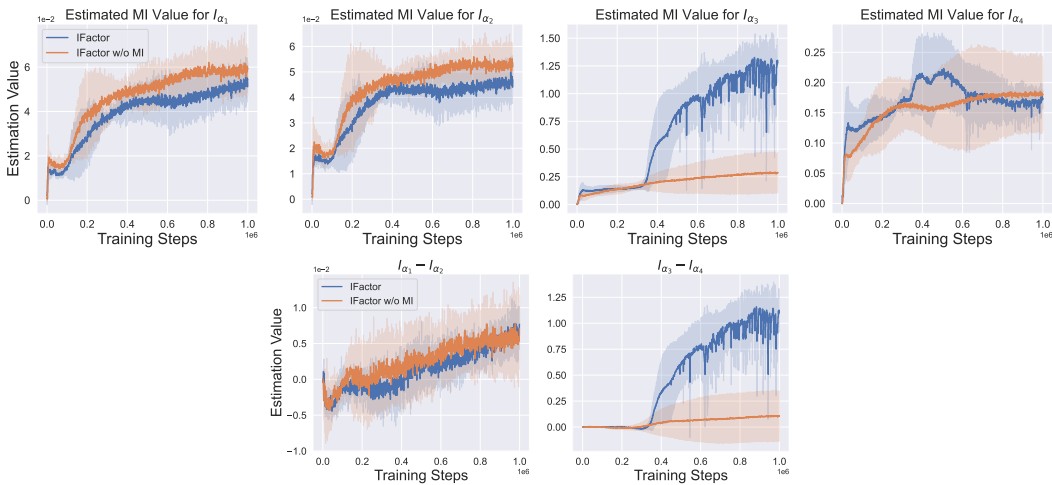

Figure 13: Estimation of the value of four mutual information terms and their differences in the Reacher Easy environment with video background and jittering camera.
.

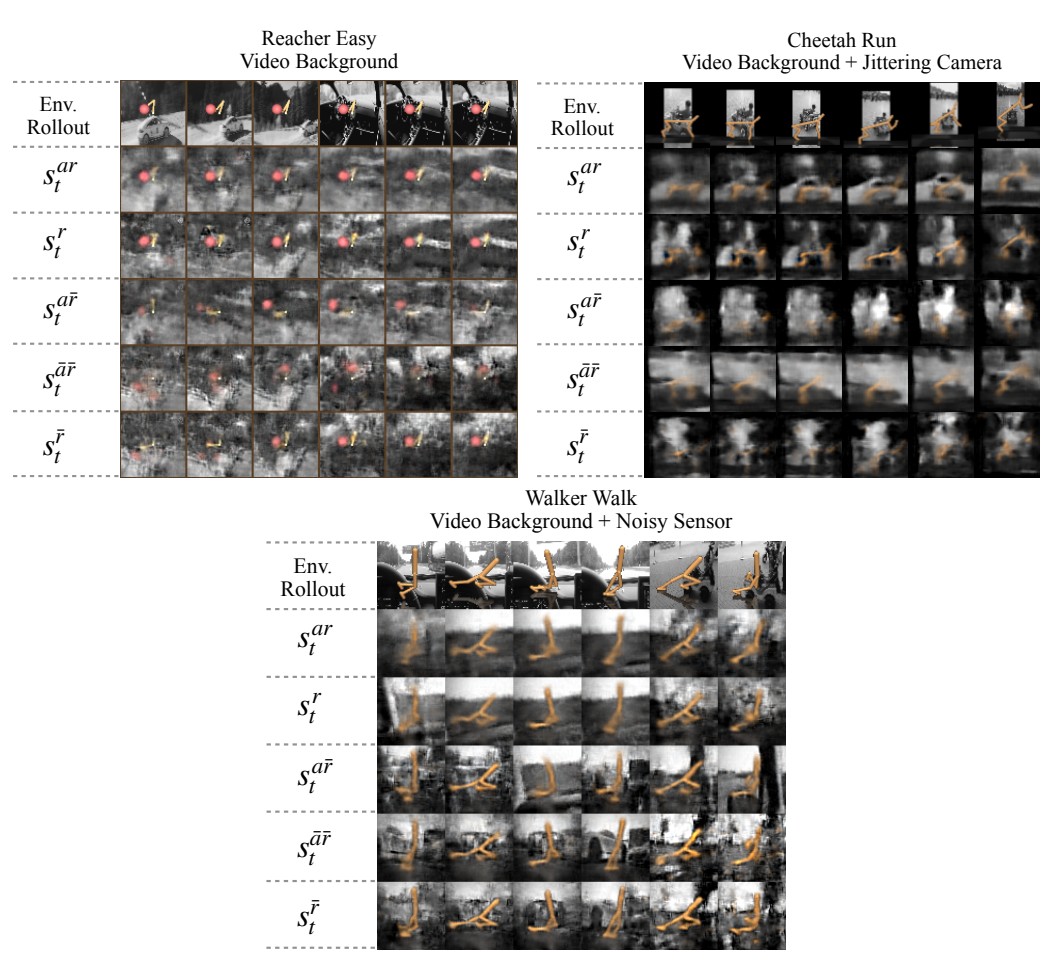

Figure 14: Visualization of the DMC variants and the factorization learned by IFactor.

## F  Comparison between IFactor and Related Work

### F.1  Comparison between IFactor and Denoised MDP

While both IFactor and Denoised MDP share the common aspect of factorizing latent variables based on controllability and reward relevance, it is crucial to recognize the numerous fundamental distinctions between them.

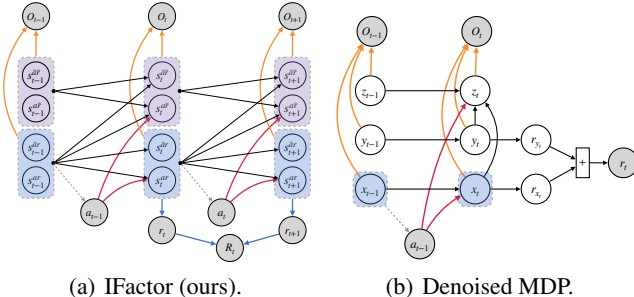

(a) IFactor (ours).          (b) Denoised MDP.

Figure 15: Graphical illustration of our world model and Denoised MDP.

First and foremost, Denoised MDP adopts a rather stringent assumption by solely considering three types of latent variables and assuming independent latent processes for $x_t$ (controllable and reward-relevant) and $y_t$ (uncontrollable but reward-relevant). However, this strict assumption may not hold in many scenarios where uncontrollable yet reward-relevant factors exhibit dependencies on controllable and reward-relevant factors. Take, for instance, the case of car driving: the agent lacks control over surrounding vehicles, yet their behaviors are indirectly influenced by the agent's actions. In contrast, our approach encompasses four types of causally related latent variables while only assuming conditional independence when conditioning on the state in the previous time step. This assumption holds true naturally within the (PO)MDP framework.

Secondly, Denoised MDP is limited to only factoring out additive rewards $r_{y_t}$ from $r_{x_t}$, disregarding the possibility of non-additive effects in many uncontrollable yet reward-relevant factors. In contrast, our method embraces the inclusion of non-additive effects of $s_t^{\bar{a}r}$ on the reward, which is more general.

Thirdly, Denoised MDP uses only controllable and reward-relevant latent variables $x_t$ for policy optimization, which we show in the theoretical analysis that it is generally insufficient. In contrast, our method utilize both controllable and uncontrollable reward-relevant factors for policy training.

Finally, Denoised MDP makes the assumption of an instantaneous causal effect from $x_t$ to $z_t$ and from $y_t$ to $z_t$, which is inherently unidentifiable without further intervention. It is worth noting that imposing interventions on the latent states is unrealistic in most control tasks, as agents can only choose actions at specific states and cannot directly intervene on the state itself. In contrast, our method assumes that there exists no instantaneous causal effect for latent variables. In conjunction with several weak assumptions, we provide a proof of block-wise identifiability for our four categories of latent variables. This property serves two important purposes: (1) it ensures the removal of reward-irrelevant factors and the utilization of minimal and sufficient reward-relevant variables for policy optimization, and (2) it provides a potential means for humans to comprehend the learned representations within the reinforcement learning (RL) framework. Through latent traversal of the four types of latent variables, humans can gain insights into the specific kind of information that each category of representation contains within the image.

From the perspective of model structure, it is worth highlighting that the architecture of both the transition model (prior) and the representation model (posterior) in IFactor differs from that of Denoised MDP. The structure of prior and posterior of IFactor is shown as follows:

$$
\begin{cases}
p_{\gamma_1}\left(s_t^{ar} \mid s_{t-1}^r, a_{t-1}\right) \\
p_{\gamma_2}\left(s_t^{\bar{a}r} \mid s_{t-1}^r\right) \\
p_{\gamma_3}\left(s_t^{a\bar{r}} \mid \mathbf{s_{t-1}}, a_{t-1}\right) \\
p_{\gamma_4}\left(s_t^{\bar{a}\bar{r}} \mid \mathbf{s_{t-1}}\right)
\end{cases}
\qquad
\begin{cases}
q_{\phi_1}\left(s_t^{ar} \mid o_t, s_{t-1}^r, a_{t-1}\right) \\
q_{\phi_2}\left(s_t^{\bar{a}r} \mid o_t, s_{t-1}^r\right) \\
q_{\phi_3}\left(s_t^{a\bar{r}} \mid o_t, \mathbf{s_{t-1}}, a_{t-1}\right) \\
q_{\phi_4}\left(s_t^{\bar{a}\bar{r}} \mid o_t, \mathbf{s_{t-1}}\right)
\end{cases}
\tag{44}
$$

Prior:          Posterior:

While Denoised MDP has the following prior and posterior:

$$
\begin{array}{ll}
\text{Prior:} & \text{Posterior:} \\
\left\{
\begin{array}{l}
p_{\gamma_1}\left(x_t \mid x_{t-1}, a_{t-1}\right) \\
p_{\gamma_2}\left(y_t \mid y_{t-1}\right) \\
p_{\gamma_3}\left(z_t \mid x_t, y_t, z_{t-1}\right)
\end{array}
\right.
&
\left\{
\begin{array}{l}
p_{\phi_1}\left(x_t \mid x_{t-1}, y_{t-1}, z_{t-1}, o_t, a_{t-1}\right) \\
p_{\phi_2}\left(y_t \mid x_{t-1}, y_{t-1}, z_{t-1}, o_t, a_{t-1}\right) \\
p_{\phi_3}\left(z_t \mid x_t, y_t, o_t, a_{t-1}\right)
\end{array}
\right.
\end{array}
\tag{45}
$$

A notable distinction can be observed between Denoised MDP and IFactor in terms of the assumptions made for the prior and posterior structures. Denoised MDP assumes independent priors for $x_t$ and $y_t$, whereas IFactor only incorporates conditional independence, utilizing $s_{t-1}^r$ as input for the transition of both $s_t^{ar}$ and $s_t^{\bar{a}r}$. Moreover, the posterior of $y_t$ receives $a_{t-1}$ as input, potentially implying controllability. Similarly, the posterior of $x_t$ incorporates $z_{t-1}$ as input, which may introduce noise from $z_{t-1}$ into $x_t$. These implementation details can deviate from the original concept. In contrast, our implementation ensures consistency between the prior and posterior, facilitating a clean disentanglement in our factored model.

From the perspective of the objective function, IFactor incorporates two supplementary mutual information constraints, namely $\mathcal{J}_{\mathrm{RS}}^t$ and $\mathcal{J}_{\mathrm{AS}}^t$, to promote disentanglement and improve policy performance.

### F.2 Comparison between IFactor, InfoPower and IsoDream

IFactor learns different categories of state representations according to their relation with action and reward, which is different from InfoPower [48] and IsoDream [12], and moreover, IFactor emphasizes block-wise identifiability for the four categories of representations while InfoPower and Iso-Dream do not. Specifically, InfoPower learns 3 types of latent variables, including $s_t^{ar}$, $s_t^{\bar{a}r}$ and $s_t^{\bar{a}\bar{r}}$. IsoDream uses three branches for latent dynamics, distinguishing controllable, noncontrollable, and static parts.

**Other differences with InfoPower:**

- **Reconstruction Basis**: InfoPower is reconstruction-free, while IFactor is reconstruction-based.
- **Objective Functions**: InfoPower prioritizes mutual information and empowerment, while IFactor utilizes reconstruction, KL constraints, and mutual information constraints for disentanglement. InfoPower formulates policy using task reward value estimates and the empowerment objective, while IFactor learns policy by maximizing the estimated Q value and dynamics backpropagating.

**Other differences with IsoDream:**

- **Objective Functions and Dynamics Modeling**: IsoDream models controllable transitions using inverse dynamics, while IFactor disentangles with multiple mutual information and KL divergence constraints. IsoDream learns policy using a future-state attention mechanism rooted in present controllable and future uncontrollable states. In contrast, IFactor focuses on reward-relevant states, ensuring $s_t^r$ variables are optimal for policy optimization.

