# OpenReview forum: "Learning World Models with Identifiable Factorization"
_NeurIPS.cc/2023/Conference — NeurIPS 2023 poster_

### Official Review · Reviewer_oFQM · 2023-07-05

**Soundness:** 3 good
**Presentation:** 3 good
**Contribution:** 2 fair
**Rating:** 6
**Confidence:** 4

**Summary:**

Learning efficient world models requires architectural priors to learn better representations that can capture different aspects of the environment. However, existing methods like Dreamer do not focus on learning disentangled representations and lack the ability to separate noise from the reward-relevant information in the observations. In this work, a new method is proposed where the latent state is divided into 4 blocks based on the dependence of action and reward function. The paper shows that using the blocks of states that affect the reward function can be used to learn policies. Experiments show the efficacy of the proposed algorithm on Robodesk and DMControl tasks with distractions and noise in the form of background or based on a sensor/camera. Moreover, ablation studies show that the model learns to disentangle the factors in the environment. Lastly, the paper also provides a theoretical justification for the proposed architecture.

**Strengths:**

1. The idea of disentangling the learned representations to different factors based on actions and rewards is interesting and nicely explained in the paper.
2. The experiments are thorough and support the claims made in the paper that the agent is able to differentiate between parts of the state that determines the reward and parts of the state that change based on the action.
3. The paper presents a theoretical derivation of the identifiability of the latent state representations.


**Weaknesses:**

1. World Models have been extended to scenarios with noisy distractions in [1][2][3]. They have shown improvements over the Dreamer method for such tasks. However, these methods are not used as baselines in the experiments.
2. Some ablations and experiments are required to further understand the scenarios where the proposed method works (More on this in Questions below)


**Questions:**

1. For learning efficient latent representations, recent works like DreamerPro have used auxiliary losses to extract relevant concepts from the observation [1]. How does the proposed method compare with reconstruction-free model-based methods?
2. For experiments on control tasks, the noise used is the background noise. There can be other types of noises as shown in [2] where distractor objects are moving in the environment. Will the proposed method be able to disentangle such noises too?
3. Were there any experiments conducted on a few games of Atari (like MsPacman, PrivateEye, and Montezuma’s Revenge) and observe what the latent embeddings are learning? On the current task, the whole agent is visible in the observation, what happens when the agent gets a partial view of the environment?
4. Related work should include [4] as they also plan to learn latent representations that disentangle different objects. Furthermore, how will the proposed method work in such scenarios where different rewards are associated with different objects?
5. Will the proposed method work in multi-task learning or meta-learning scenarios where the rewards are changing because there is a part of latent space that depends on the reward function which may change with different reward functions?
6. The current method assumes the presence of a reward function. Will this assumption make it difficult to learn unsupervised world models? Here the latent space cannot be conditioned on the intrinsic reward as it changes with training.

### References
[1] Deng, Fei, Ingook Jang, and Sungjin Ahn. "Dreamerpro: Reconstruction-free model-based reinforcement learning with prototypical representations." International Conference on Machine Learning. PMLR, 2022.\
[2] Jain, Arnav Kumar, et al. "Learning robust dynamics through variational sparse gating." Advances in Neural Information Processing Systems 35 (2022): 1612-1626.\
[3] Nguyen, Tung D., et al. "Temporal predictive coding for model-based planning in latent space." International Conference on Machine Learning. PMLR, 2021.\
[4] Kipf, Thomas, Elise Van der Pol, and Max Welling. "Contrastive learning of structured world models." arXiv preprint arXiv:1911.12247 (2019).


**Limitations:**

The paper briefly talks about a few limitations between lines 346-350 and needs to elaborate more on the discussion and why they are challenging.

---

> ### Author Rebuttal · Authors · 2023-08-10
>
> **Dear Reviewer,**
>
> Thank you for your valuable feedback. Below, we address your concerns in a point-by-point manner and have added various experiments, following your suggestions.
>
> **Weaknesses:**
>
> 1. Lack recent works in world models such as [4,5,6] as baselines
>
>     *Response:*  Thank you for bringing our attention to the related works. In our updated experiments, we have included DreamerPro [4] as a baseline for DMC variants. The results, available in the general response and newly uploaded pdf file, consistently show IFactor outperforming DreamerPro. Variational Sparse Gating (VSG, [5]) introduces a distinct latent dynamics model that deviates from the typical RSSM structure. We have not included [5] in our baseline comparisons because IFactor and VSG actually address different aspects, and we are currently working on combining them for potential performance improvement. Moreover, [6] proposes an alternative approach using temporal predictive coding, where latent states may not be identifiable due to the non-invertibility of the mixing function f. We have not included [6] in our baselines because its implementation isn't yet publicly available.
>
> 2. Requirements for experiments to further understand the scenarios where the proposed method works.
>
>     *Response:*  Thank you for your suggestions. We offer detailed responses to each question below.
>
>
> **Questions:**
>
> **Q1:**  Comparative Analysis with reconstruction-free model-based methods, such as DreamerPro:
>
> **A1:** We've expanded our experiments to incorporate DreamerPro as a baseline for DMC variants. Results are shown at the end of the general response and in Figure 4 of the newly uploaded pdf file. Notably, IFactor consistently surpasses DreamerPro's performance.
>
> **Q2:** Disentanglement of Noises such as  distractor objects in the environment:
>
> **A2:** Thank you for your insightful thoughts. We have indeed addressed the concept of distractor objects moving within the environment in Section 5.1.2, where we considered a modified Cartpole environment with a distracting cart that is neither reward-relevant nor action controllable. The identifiability score ($R^2$) is given in Figure 2 of the newly uploaded pdf file. Following your suggestions, we are now working on the environment presented in [5].
>
> **Q3:** Experiments on Atari games and Partial-view Environments:
>
> **A3:** Thank you! Following your comments, we are now testing our approach to Atari games. Considering the achievements of Dreamer  (v2/v3) in similar situations and the flexibility of our method to work with different representation types and training techniques, we believe that combining IFactor with Dreamer (v2/v3)  holds the potential. Our visualizations in Figures 4 and 5 of the main paper, and Figures 1 and 6 in the appendix, offer insights into how latent features are organized.
>
> **Q4:**  Inclusion of Related Work [7] and performance of IFactor in scenarios where different rewards are associated with different objects:
>
> **A4:** Thank you for bringing the related work [7] to our attention. We will make sure to include the proper citation in our revised version. Moreover, When dealing with multiple rewards, our proposed framework actually can be directly extended to identify the state representation or object that is associated with each reward signal. To illustrate, suppose there are two reward signals r1 and r2 that are associated with object 1 and object 2, respectively. Then with our framework, we can recover the representation $s^{r1}$ that is relevant to reward r1 and $s^{r2}$ that is relevant to reward r2.
>
> **Q5:** Functionality in Multi-task or Meta-learning:
>
> **A5:** Yes! In fact, our framework can be easily extended to cover heterogeneous/nonstationary environments, with the change of reward function, observation function, or transition dynamics. Basically, we can introduce a (latent) low dimensional change factor to characterize those changes, a similar strategy as AdaRL. This is the direction we plan to take in our next steps.
>
> *Reference:* *[AdaRL] Huang, Feng, Lu, Magliacane, Zhang. AdaRL: What, Where, and How to Adapt in Transfer Reinforcement Learning. ICLR, 2022.*
>
> **Q6** Application in Unsupervised World Models:
>
> **A6:**  Our framework is versatile when it comes to different supervision signals. Utilizing the reward and action signals in reinforcement learning systems is merely one specific application of its capabilities. In purely unsupervised world models without any supervision signals, we are unable to distinguish different categories of state representations. However, if intrinsic rewards are present, we can learn the state representation relevant to these intrinsic rewards.
>
> **Limitations:**
>
> In-depth discussion: While recent studies suggest that causal variables can be reconstructed from temporal sequences of observations, assuming no instantaneous causal relations, practical challenges arise. Specifically, if our measurement or frame rate lags behind the speed of causal effects, it can inadvertently introduce “instantaneous” effects, compromising prior identifiability conclusions. Our work's foundational limitation rests on the assumption that latent processes lack instantaneous causal relations. This assumption is evident in Figure 1(c) where no edges exist between concurrent latent states. For a deeper dive, we direct readers to two pivotal papers: [8] introduces iCITRIS, a method allowing instantaneous effects in intervened temporal sequences when intervention targets are observable. Meanwhile, [9] articulates that neglecting instantaneous dependence can lead to subpar policy learning in MBRL. We commit to elaborating on these limitations in the camera-ready edition of our paper.
>
> Thank you once again for your rigorous review and constructive feedback.
>
> The reference index can be found at the end of the general response.

---

> > ### Comment · Reviewer_oFQM · 2023-08-13
> >
> > Thank you for the detailed response. Regarding A6: Intrinsic rewards are non-stationary, will the method adapt to changing rewards? For the same state, the reward will change with time depending on how many times the state has been visited.
> >
> > Overall, I am convinced with the response and hope the authors will make the changes to the paper. I have updated my score.

---

> > > ### Author Response · Authors · 2023-08-14
> > > **Thank you very much for carefully reviewing our response**
> > >
> > > Thank you very much for carefully reviewing our response and promptly updating us. Your attention is highly appreciated! In regard to the intrinsic reward scenario you proposed, one potential approach could be to include the number of times the state has been visited in the reward function. This may help capture the type of nonstationarity you mentioned.
> > >
> > > Thank you!

---

> > > > ### Comment · Reviewer_oFQM · 2023-08-14
> > > >
> > > > I understand that the count of states can account for uncertainty. But based on this intrinsic reward- the state representation can be factorized into components that depend on reward and do not depend on reward. As the reward is changing, my question is more about whether the learned state embedding will adapt to these changing rewards.

---

> > > > > ### Author Response · Authors · 2023-08-20
> > > > >
> > > > > Thank you once again for your thoughtful insights. We truly appreciate it. Below, we carefully check various cases when the reward function is nonstationary.
> > > > >
> > > > > 1) Case 1: If changes in the reward function can be captured by certain change factors, represented as $r_{t+1} = f(s_t, a_t, c_t, \eta_t)$, where $c_t$ denotes change factors (which could be unobserved) that vary across conditions or evolve smoothly over time, and $\eta_t$ represents stationary random noise if presents. In this situation, our approach can readily adapt to recover the reward-related state representations $s_t$ by involving $c_t$, as the shifts in distribution are encapsulated by $c_t$. This scenario has also been discussed in AdaRL.
> > > > > 2) Case 2: For general nonstationary reward functions, such as $r_{t+1}=f_t (s_t, a_t, \eta_t)$, where distribution shifts cannot be characterized by $c_t$, the performance would be influenced by the non-stationarity of the reward function. In such cases, achieving stable disentanglement becomes unattainable, making IFactor potentially unsuitable. A particular instance, as you mentioned, is that the reward is a function of the number of times the state has been visited.
> > > > >
> > > > > We will include the discussions about these cases in the final section and are investigating IFactor's performance in the face of varying degrees of reward function non-stationarity. Your profound insights and valuable comments are greatly appreciated. Thank you very much!

---

### Official Review · Reviewer_qxcU · 2023-07-06

**Soundness:** 3 good
**Presentation:** 3 good
**Contribution:** 2 fair
**Rating:** 6
**Confidence:** 3

**Summary:**

- The paper introduces a framework called IFactor for modeling latent state variables in reinforcement learning (RL) systems. The framework categorizes these variables into four distinct types based on their interactions with actions and rewards. The paper further establishes block-wise identifiability of these latent variables, which provides a stable and compact representation and discloses that all reward-relevant factors are significant for policy learning. Overall, the paper contributes a comprehensive framework, IFactor, that provides representations of different aspects of information in RL systems and highlights the importance of considering states that influence the reward during decision-making.

**Strengths:**

- IFactor models different latent state variables in based on their interactions with actions and rewards in RL systems. These categories include reward-relevant and controllable parts, reward-irrelevant but uncontrollable parts, controllable but reward-irrelevant parts, and unrelated noise.
- The paper defines blockwise identifiability as the existence of a one-to-one mapping between a latent variable and its estimated value, such that the estimated value can be recovered from the latent variable and vice versa. This means that each latent variable can be uniquely identified and separated from other variables in the model, allowing for a more stable and compact representation of the environment.
- The paper includes extensive experiments and ablation studies to validate the effectiveness of the proposed framework and to analyze the impact of different components. Meanwhile, the paper includes a thorough evaluation of the proposed framework on several benchmark tasks.

**Weaknesses:**

- Missing related work. The separate modeling of controllable and uncontrollable components has been explored in various previous works, such as InfoPower[1] and Iso-Dream[2], which are missing from related work. It would be better if the authors can discuss the differences between the proposed method and this type of work.
  - [1] INFOrmation Prioritization through EmPOWERment in Visual Model-Based RL. ICLR 2022.
  - [2] Iso-Dream: Isolating and Leveraging Noncontrollable Visual Dynamics in World Models. NeurIPS 2022.
- The error bars of the baselines are not included.
- The performance of TIA on variants of DMC sometimes encounters a sudden drop during training, which is worth a further check for the reason.
- Except for DenoisedMDP, the compared approaches are somewhat "out-of-date". It would be nice if the authors could include stronger baseline methods for model comparison, especially the approaches that show impressive performance on generalization in noisy visual observations.

**Questions:**

- I understand that blockwise identification refers to the existence of a one to one mapping between a late variable and its estimated value. However, I lacked a thorough understanding of how decoupling the four latent components is related to the concept of block-wise identifiability. Could you please explain further? Or what are the benefits of defining these components blockwise identifiable?
- Additionally, if this mapping is not blockwise identifiable, what impact will it have on the decoupling performance and how will it influence the model performance?

**Limitations:**

Negative societal impact is not discussed

---

> ### Author Rebuttal · Authors · 2023-08-10
>
> Dear Reviewer,
>
> Thank you for your valuable feedback. Please see our responses to your questions point-by-point below.
>
> 1. M**issing related work on separate modeling**:
>
>     Thank you for bringing our attention to the related works, including InfoPower [1] and Iso-Dream [2]. We will make sure to give proper citations and provide comparisons in our appendix for the revised version. A brief discussion about differences is provided below.
>
>     Main differences: IFactor learns different categories of state representations according to their relation with action and reward, which is different from InfoPower and IsoDream, and moreover, IFactor emphasizes block-wise identifiability for the four categories of representations while InfoPower and Iso-Dream do not. Specifically, InfoPower learns 3 types of latent variables, including  $s^{ar}_t, s^{\bar{a}r}_t$ and  $s^{\bar{a}\bar{r}}_t$. IsoDream uses three branches for latent dynamics, distinguishing controllable, noncontrollable, and static parts.
>
>     - Other differences with InfoPower:
>         - **Reconstruction Basis**: InfoPower is reconstruction-free, while IFactor is reconstruction-based.
>         - **Objective Functions**:  InfoPower prioritizes mutual information and empowerment, while IFactor utilizes reconstruction, KL constraints, and mutual information constraints for disentanglement. InfoPower formulates policy using task reward value estimates and the empowerment objective, while IFactor learns policy by maximizing the estimated Q value and dynamics backpropagating.
>     - Other differences with IsoDream:
>         - **Objective Functions and Dynamics Modeling**: IsoDream models controllable transitions using inverse dynamics, while IFactor disentangles with multiple mutual information and KL divergence constraints. IsoDream learns policy using a future-state attention mechanism rooted in present controllable and future uncontrollable states. In contrast, IFactor focuses on reward-relevant states, ensuring $s^r_t$ variables are optimal for policy optimization.
> 2. **Absence of error bars for baselines**:
>
> Thank you for your careful checking. We didn't include error bars for baselines due to the following reasons. Initially, we tried to replicate the results of Denoised MDP [3] using their available code. However, the outcomes we obtained were considerably poorer than those reported in their paper (refer to Figure 4 in the appendix). In order to maintain a fair comparison, we directly incorporated the reported results from Denoised MDP's image plots. The absence of error bars is because we were unable to extract them from these image plots. In light of your feedback, we have made the subsequent changes.
>
> 1. We have included error bars for Figure 3 in the newly uploaded pdf file. They will be included in the final version.
> 2. We have also listed the mean and standard deviation of the return value for all baselines, when policy optimization gets converged, at the end of the general response (these values are directly copied from the reported results). These tables will also be added to the appendix of the revised manuscript.
>
> 1. **Performance inconsistencies of TIA**:
>
>     To maintain fair comparisons, the results of TIA were directly copied from Denoised MDP. We will replicate and provide detailed explanations of TIA's results in our revised manuscript.
>
> 2. **Comparison with contemporary baselines**:
>
>     Thank you for the suggestion. Considering the continuous progress in RL, we've added results from DreamerPro, a top model-based RL method that doesn't require reconstruction and can handle noisy visuals effectively, as a baseline. You can find the updated results in Figure 4 of the newly uploaded pdf file and at the end of the general reponse, where IFactor achieves superior performance.
>
>
> ## **Questions:**
>
> Q1: Block-wise Identifiability & Decoupling:
>
> A1: Basically, block-wise identifiability ensures the theoretical guarantee of the estimations of the four categories in relation to their ground truth values. With identifiability, we can ensure asymptotic correctness. Moreover, it is important to note that decoupling alone does not guarantee identifiability. Specifically, solely relying on decoupling may lead to missing reward-relevant information in the estimation $\hat{s}^r_t$, as well as the presence of redundant information.
>
> Q2: Benefits of block-wise identifiability and consequences of non-blockwise identifiability
>
> A2: Since block-identifiability guarantees the recovery of ground truth variables, it additionally offers the advantages of improved interpretability and facilitating more efficient and effective policy optimization, which are also the downsides of missing identifiability.
>
> - **Interpretability**: It guarantees the recovery of underlying causal latent variables under certain conditions, thereby enhancing the clarity of decision-making processes by distinguishing latent state variables based on controllability and reward relevance.
> - **Policy Optimization**: By focusing on $s^r_t$, IFactor streamlines policy learning, keeping only essential information and discarding redundancy. If the latent variables lack block-wise identifiability, $s^r_t$ may have redundant or insufficient information, hindering efficient and effective policy learning.
>
> We deeply appreciate the depth and breadth of your review. Your insights are crucial in improving our manuscript. We will integrate your feedback into our revised version.
>
> The reference index can be found at the end of the general response.

---

> > ### Comment · Reviewer_qxcU · 2023-08-20
> >
> > Thank you for the detailed explanations which have clarified my concerns. I am willing to raise my score.

---

> > > ### Author Response · Authors · 2023-08-20
> > >
> > > Thank you for your thorough review and subsequent feedback on our response. We truly value your insights as they significantly contribute to the improvement of our paper. Thank you!

---

> ### Author Response · Authors · 2023-08-20
>
> Thank you for your insightful review of our paper. We've diligently addressed the feedback in our rebuttal and observed updates from two other reviewers. Could you kindly let us know if our response has addressed your concerns? If any issues remain, please don't hesitate to inform us. We'd appreciate it if you could review our rebuttal and update your feedback or score when convenient. Your time and effort are greatly valued.

---

### Official Review · Reviewer_gMe8 · 2023-07-06

**Soundness:** 3 good
**Presentation:** 3 good
**Contribution:** 3 good
**Rating:** 7
**Confidence:** 3

**Summary:**

The authors propose an alternative way to create world models in an RL system by separating them into blocks dependant on their casual effects on future observations, states and rewards. They theoretically show that under some assumptions it is possible to identify such classes of variables, even if specific instances cannot be identified. They go on to show how their approach can increase robustness and disentanglement in several control tasks.

**Strengths:**

1. The proposed method is well motivated. I especially liked that instead of focusing on variable level disentanglement the authors have opted for block level disentanglement. This is a good way to relax the assumptions of the model without being completely unconstrained.
2. The theoretical motivation seems good (but see below). It is always a bonus when researchers can provide some theoretical justification for their work.
3. Experimental evidence is convincing. I especially liked the analysis of the disentanglement of structured datasets and the results look extremely promising.

**Weaknesses:**

No serious weakness. I have some comments regarding how section 2 and 3 are written but otherwise it is a solid paper.

**Questions:**

In general my questions revolve around the quality of the plots and some results that I think are missing.

1. I would like to sees disentanglement scores for the comparison models in Figure 3. While it is true that these do not posses explicit block level disentanglement, this could potentially emerge from training. I personally don't believe this is likely, but it is still important that the authors show that it is not the case.
2. Why are there no disentanglement scores for the cart pole dataset? The plot is very small too. This needs must be larger otherwise it is not easy to read. Figure 5 has the same issue.

**Limitations:**

1. Section 2 could use a slight rewrite. Each of the propositions in lines 98 to 113 start the same way. Seems like it should be possible to just use one heading and enumerate the propositions afterward.
2. I would prefer if the authors gave more intuitions in section 3 and left the theorem for the appendix. I find that when reading articles it's much easier to reason about the approaches if a good intuition is given than if I am presented with a technical derivation of a theorem. Speaking of which it is not clear why property A2 in Theorem 1 is needed. Also, I cannot find what the definition of $\hat{s}$ in the text until it is used in Definition 1.

---

> ### Author Rebuttal · Authors · 2023-08-10
>
> **Dear Reviewer,**
>
> Thank you for your detailed feedback on our paper. Your helpful comments have helped us further improve the paper. We've worked hard to deal with all the things you pointed out, and here, we explain the changes we made based on your advice.
>
> **Q1: Disentanglement scores for comparison models in Figure 3**
>
> *Response:* Thanks for your comments. To clarify, we provided averaged disentanglement (or, as we prefer, 'identifiability') scores ($R^2$) in the right two panels of Figure 3, which are derived as the mean of the four diagonal values. This was intended to show how the identifiability scores change during training. Following your feedback, we have now computed and appended the identifiability scores for the optimal model iteration for all comparison models. You can find the results in the newly uploaded pdf file, and they will also be included in the appendix of the final version.
>
> **Q2: Disentanglement scores for the cart pole dataset and legibility issues in Figures**
>
> *Response:* Thanks for your careful reading. We have included the identifiability scores for the cart pole dataset in Figure 2 of the newly uploaded pdf file. Unlike the synthetic dataset, where there are clear categories for latent state variables, real-world situations pose a challenge due to the potential ambiguity in categorizing true variables. In our modified Cartpole environment, we defined the ground-truth $s^{ar}_t$ as the cart's position, $s^{\bar{a}r}_t$ as the pole's angle, $s^{a\bar{r}}_t$ as the light's greenness, and $s^{\bar{a}\bar{r}}_t$ as the distractor Cartpole's state (including cart position and pole angle). The disentanglement scores for individual $s^{ar}_t$ and $s^{\bar{a}r}_t$, as well as the combined $s^r_t$, are shown in Figure 2 of the supplementary file. We can obviously see that the true latent variables can be clearly identified. Additionally, we have enlarged Figures 4 and 5 in the revised version to improve readability.
>
> **Limitations:**
>
> 1. **Structure of Propositions**
>
> *Response:* We really appreciate your careful observation regarding the repetitive structure between lines 98 and 113. In line with your suggestion, we will simplify this section by consolidating it under a single heading and providing numerical listings for the propositions. This modification will be present in the paper's final version.
>
> 1. **Intuition in Section 3 and Clarifications on Theorem 1**
>
> Thanks for your great suggestion. Intuitively, property A2 in Theorem 1 requires that the Jacobian varies “enough” so that it cannot be contained in a proper subspace of R. This requirement is necessary to avoid undesirable situations where the problem becomes ill-posed and is essential for identifiability. A special case when this property does not hold is when the function f is linear, as the Jacobian remains constant in such cases. We will give a more detailed explanation in the main text, accompanied by a more intuitive outline of the proof.
>
> We sincerely hope our revisions and clarifications align with your expectations. Your invaluable insights have definitely improved the quality of our research.

---

> > ### Comment · Reviewer_gMe8 · 2023-08-14
> >
> > I thank the authors for their response. My concerns have been addressed and I look forward to their updated version with the proposed changes.

---

> > > ### Author Response · Authors · 2023-08-15
> > >
> > > Thank you very much for carefully reviewing our response. We really appreciate your feedback and will make sure to include the changes into our updated version.
> > >
> > > Thank you!

---

### Author Rebuttal · Authors · 2023-08-10

Dear Reviewers,

We sincerely thank the reviewers for their effort and helpful comments regarding our paper. We have carefully revised the manuscript according to your comments.  A summary of the primary changes we've made is outlined below:

1. **Identifiability Scores**: We've incorporated Identifiability Scores ($R^2$) for the compared models in the synthetic dataset experiments, as well as Identifiability Scores for IFactor in the modified Cartpole environment. Both can be referenced in the newly uploaded pdf file.
2. **Structural Changes**: Propositions from lines 98 to 113 have been organized under a singular heading, followed by an enumeration for improved clarity.
3. **Theoretical Clarification**: An intuitive elaboration for Assumption 2 within Theorem 1 has been added.
4. **Literature Citations**: Proper attributions to related works, namely InfoPower [1], Iso-Dream [2], DreamerPro [4], VSG [5], TPC [6] and C-SWMs [7] have been integrated. We ensure a thorough comparison with these in the revised manuscript.
5. **Experimental Enhancements**:
    - DreamerPro is now incorporated as an additional baseline in the DMC variants experiments. The results can be found in the tables at the end of this response and in Figure 4 of the newly uploaded pdf file.
    - The middle panel of Figure 5 now displays error bars for baselines.
    - For a comprehensive understanding, three tables elucidating the mean and standard deviation values of the performance of the converged policy in DMC variants experiments have been appended.
6. **Discussion Expansion**: A more in-depth discussion on the limitations between lines 346-350 is now available.

To get a thorough look at these changes, we direct your attention to the newly uploaded pdf file and the specific responses to the reviewers.

We hope our revisions and detailed responses satisfactorily address the concerns raised. Once again, thank you for your generous contribution of time and expertise to the community.

## Evaluation of Policy Performance in DMC Variants
**cheetah**

|  | IFactor (Ours) | DreamerPro | Denoised MDP | Dreamer | TIA | DBC | CURL | PI-SAC |
| --- | --- | --- | --- | --- | --- | --- | --- | --- |
| Noiseless | **874.0±39.0** | 802.5±75.0 | 770.7±59.9 | 713.7±347.8 | 765.9±29.3 | 181.8±49.7 | 171.2±34.2 | 369.4±27.7 |
| Video Background | **572.5±192.5** | 366.0±96.4 | 430.6±111.0 | 171.0±44.9 | 387.8±297.3 | 247.9±71.3 | 70.3±47.6 | 111.9±31.0 |
| Noisy Sensor | **455.5±87.9** | 134.3±41.1 | 400.3±190.0 | 166.0±41.7 | 226.8±13.2 | 140.6±13.0 | 198.9±7.1 | 150.9±14.1 |
| Camera Jittering | **417.7±22.4** | 150.0±104.4 | 293.9±99.6 | 160.1±32.2 | 202.1±93.2 | 141.3±35.4 | 168.6±21.6 | 155.8±19.6 |

**walker**

|  | IFactor (Ours) | DreamerPro | Denoised MDP | Dreamer | TIA | DBC | CURL | PI-SAC |
| --- | --- | --- | --- | --- | --- | --- | --- | --- |
| Noiseless | **966.0±4.5** | 941.4±16.4 | 947.4±13.3 | 954.9±6.0 | 955.3±5.2 | 613.9±110.7 | 416.5±295.8 | 202.9±92.0 |
| Video Background | **916.8±51.7** | 909.1±48.5 | 790.2±113.3 | 247.2±134.6 | 685.4±336.6 | 198.6±67.3 | 607.8±99.7 | 200.3±18.2 |
| Noisy Sensor | **700.5±174.3** | 242.1±64.9 | 660.7±120.1 | 269.6±145.0 | 424.7±281.2 | 95.1±53.7 | 338.4±91.7 | 221.9±20.8 |
| Camera Jittering | **523.7±194.2** | 367.8±300.9 | 290.5±103.9 | 105.9±22.1 | 229.5±331.9 | 62.0±17.3 | 447.5±69.5 | 115.6±5.9 |

**Reacher**

|  | IFactor (Ours) | DreamerPro | Denoised MDP | Dreamer | TIA | DBC | CURL | PI-SAC |
| --- | --- | --- | --- | --- | --- | --- | --- | --- |
| Noiseless | 923.7±37.2 | **924.2±61.2** | 685.7±216.4 | 876.1±57.1 | 587.1±256.3 | 95.2±57.5 | 663.4±220.5 | 166.2±235.4 |
| Video Background | **962.7±9.5** | 555.1±91.5 | 543.9±121.3 | 252.8±127.0 | 123.2±21.3 | 101.6±57.7 | 751.1±188.9 | 76.2±34.6 |
| Noisy Sensor | **839.3±50.9** | 675.1±137.2 | 561.1±182.4 | 201.8±81.6 | 263.7±279.8 | 96.8±38.6 | 606.8±259.7 | 84.7±4.5 |
| Camera Jittering | **735.7±52.6** | 674.5±81.6 | 213.1±105.9 | 108.9±18.9 | 89.3±25.7 | 86.7±50.6 | 631.9±96.0 | 84.3±13.1 |

[1] Bharadhwaj, Homanga, et al. "Information prioritization through empowerment in visual model-based rl.” ICLR 2022.

[2] Pan, Minting, et al. "Iso-dream: Isolating and leveraging noncontrollable visual dynamics in world models." *Advances in Neural Information Processing Systems* 35 (2022): 23178-23191.

[3] Wang, Tongzhou, et al. "Denoised mdps: Learning world models better than the world itself." ICML 2022: 22591-22612

[4] Deng, Fei, Ingook Jang, and Sungjin Ahn. "Dreamerpro: Reconstruction-free model-based reinforcement learning with prototypical representations." International Conference on Machine Learning. PMLR, 2022.

[5] Jain, Arnav Kumar, et al. "Learning robust dynamics through variational sparse gating." Advances in Neural Information Processing Systems 35 (2022): 1612-1626.

[6] Nguyen, Tung D., et al. "Temporal predictive coding for model-based planning in latent space." International Conference on Machine Learning. PMLR, 2021.

[7]  Kipf, Thomas, Elise Van der Pol, and Max Welling. "Contrastive learning of structured world models." arXiv preprint arXiv:1911.12247 (2019).

[8] Lippe, Phillip, et al. "Causal representation learning for instantaneous and temporal effects in interactive systems." *The Eleventh International Conference on Learning Representations*. 2022.

[9] Zhu, Zhengmao, et al. "Beware of Instantaneous Dependence in Reinforcement Learning." *arXiv preprint arXiv:2303.05458*(2023).

---

### Decision · Program_Chairs · 2023-09-21

**Decision:**

Accept (poster)

**Comment:**

The authors proposed to learn world models with additional structural assumptions on the causal graph. Specifically, the authors assumed that state variables can be decomposed into four parts with the reward-relevancy and controllability, and discussed when and how these four parts can be identified from the data. Experimental results demonstrate the benefits of such structural assumptions in DMC tasks. Reviewers all agree that the proposed methods are well-motivated and properly justified. I encourage the authors to incorporate the suggestions from reviewers into the final version.